# WHEN LLMs PLAY THE TELEPHONE GAME: CULTURAL ATTRACTORS AS CONCEPTUAL TOOLS TO EVALUATE LLMs IN MULTI-TURN SETTINGS

**Jérémy Perez**[*,1]**, Grgur Kovač**[1]**, Corentin Léger**[1]**, Cédric Colas**[1,2]**, Gaia Molinaro**[1,3]**, Maxime Derex**[4]**, Pierre-Yves Oudeyer**[1]**, and Clément Moulin-Frier**[1]

[1]Inria, Flowers team, Université de Bordeaux, France
[2]MIT, Computational Cognitive Science Lab, Cambridge, MA, USA
[3]Department of Psychology, University of California, Berkeley, Berkeley, CA, USA
[4] Institute for Advanced Study in Toulouse, Toulouse, France

## ABSTRACT

As large language models (LLMs) start interacting with each other and generating an increasing amount of text online, it becomes crucial to better understand how information is transformed as it passes from one LLM to the next. While significant research has examined individual LLM behaviors, existing studies have largely overlooked the collective behaviors and information distortions arising from iterated LLM interactions. Small biases, negligible at the single output level, risk being amplified in iterated interactions, potentially leading the content to evolve towards attractor states. In a series of *telephone game experiments*, we apply a transmission chain design borrowed from the human cultural evolution literature: LLM agents iteratively receive, produce, and transmit texts from the previous to the next agent in the chain. By tracking the evolution of text *toxicity*, *positivity*, *difficulty*, and *length* across transmission chains, we uncover the existence of biases and attractors, and study their dependence on the initial text, the instructions, language model, and model size. For instance, we find that more open-ended instructions lead to stronger attraction effects compared to more constrained tasks. We also find that different text properties display different sensitivity to attraction effects, with *toxicity* leading to stronger attractors than *length*. These findings highlight the importance of accounting for multi-step transmission dynamics and represent a first step towards a more comprehensive understanding of LLM cultural dynamics.

## 1 INTRODUCTION

Large language models (LLMs) are playing an increasingly significant role in the production of media content across various domains (Brinkmann et al., 2023). They are being used for academic writing (Buruk, 2023; Agarwal et al., 2024; Cheng et al., 2024; Dergaa et al., 2023; Khraisha et al., 2024; Marlow and Wood, 2021; Cabanac et al., 2021), journalism (Petridis et al., 2023), story generation (Valentini et al., 2023; Xie et al., 2023; Zhao et al., 2023), as chatbots on social media (Sadikoğlu et al., 2023) and in the workplace at large (Brynjolfsson et al., 2023; Eloundou et al., 2023). As LLMs become more performant, controllable, and more widespread, their impact on the creation and dissemination of information content humans consume is expected to grow even further (Brinkmann et al., 2023).

Given the implications of LLM usage for the production of cultural information, numerous researchers have studied the properties of LLM-generated content. In these studies, LLMs showed several biases with respect to gender (Acerbi and Stubbersfield, 2023; Salewski et al., 2024; Wan et al., 2023; Haller et al., 2023; Dong et al., 2024), race (Raj et al., 2024), values (Atari et al., 2023; Sivaprasad et al., 2024), politics (Motoki et al., 2023; Agiza et al., 2024; Haller et al., 2023), authority, fallacy oversight,

---

[*]Corresponding author: `jeremy.perez@inria.fr`

Figure 1: **The transmission chain experimental design.** (a) Single-turn transmission: an LLM agent receives a human-generated input text (e.g. a story) and a task (e.g. "rephrase the text") and generates an output text. (b) Multi-turn transmission: a chain of LLM agents is given the same task, with the first agent receiving an initial text and subsequent agents receiving the output of the preceding agent. Measures of *toxicity*, *positivity*, *difficulty*, and *length* are recorded at each step of the chain.

and beauty (Chen et al., 2024). They were also found to generate at least as attractive (Marlow and Wood, 2021) and compelling (Spitale et al., 2023) texts than humans and to display similar cognitive biases (Echterhoff et al., 2024).

While *single-turn behaviors* of LLMs prompted with a *human-generated prompt* are under active scrutiny, little is known about the effect of iterated interactions. Indeed, LLMs often use existing cultural content to generate new ones, for example when writing scientific reviews (Khraisha et al., 2024; Agarwal et al., 2024) or generating new stories based on existing examples (Xie et al., 2023). As the share of AI-generated content increases, LLMs will be producing outputs using content that is already LLM-generated, making it crucial to study the consequence of this iterative process. Moreover, LLMs are increasingly being used in multi-agent settings (de Zarzà et al., 2023; Park et al., 2023; 2022; Xiao et al., 2023; Hua et al., 2023; Vezhnevets et al., 2023; Chuang et al., 2023) and are already interacting with one another as chatbots on social media[1]. Nonetheless, little is known about how populations of LLMs might self-organize, as other complex systems do. Research in complex systems traditionally studies how global-level patterns emerge from local interactions (Gleick and Hilborn, 1988; Mitchell, 2009). Here, we ask whether multi-turn behaviors conditioned on LLM-generated content cause the appearance of new kinds of biases, undetectable in single-turn behaviors but accumulating across iterated interactions.

To address this question, we take inspiration from the cultural evolution literature. The field of cultural evolution aims to provide causal explanations for the change of culture over time. By *cultural*, we refer to the definition traditionally used in the cultural evolution literature, where it refers to *any type of socially transmitted information* (Mesoudi, 2016). This definition includes various types of cultural information (e.g. beliefs, skills, norms) and is independent of the medium of transmission (e.g. bodily imitation, oral or written transmission). Importantly, this way of defining culture is quite different from the view that is adopted by research works studying cultural values, or cultural alignment in LLMs, where culture is seen as an attribute of a specific human group (e.g. "Western Culture"). This definition allows to use the same conceptual and methodological framework which was applied to study human (Derex et al., 2019) and non-human (Whiten et al., 2011) cultural evolution to study LLMs, which is why we adopt that definition for this work.

In particular, we draw insights from a research tradition called *cultural attraction theory* (CAT) (Sperber, 1985; Morin, 2016; Miton, 2024). CAT aims to determine how non-random transformations of cultural information during transmission events may lead to the evolution of progressively more stable forms, referred to as *attractors*. Although the precise conceptualization of attractors varies across authors, an encompassing definition may be *"theoretical posits that capture the way in which certain ideational variants are more likely to be the outcome of transformations than others."* (Buskell, 2017). Importantly, the existence of attractors would predict that for a broad range of starting conditions, different variants of a given cultural trait will converge towards the same attractor, that is, to a cultural trait that possesses specific properties.

Most experiments in CAT employ a transmission chain design (Mesoudi, 2021), first introduced by (Bartlett, 1932). In this setup, chains of participants receive, produce and transmit social information from and to each other in a sequential manner (as in the popular *telephone game*). This powerful and

---

[1]https://chirper.ai/

highly controlled design allows to evaluate the high-level patterns that emerge from the accumulation of directional changes during single-turn transmission events. Here, we adapt this design to study how culture evolves along chains of LLMs, rather than human participants (Figure 1).

We conducted several transmission chain experiments with LLMs, where the first LLM-based agent in the chain receives a human-generated text, elaborates on it, and then passes it to the next agent in the chain. This transmission step is repeated with different instances of the LLM agent until the end of the chain is reached. We simulated repeated transmission to observe the direction in which text properties evolve, starting from a broad range of initial values. Finding that certain cultural variants (for example, texts with low *toxicity*) are more likely to be the outcome of this cultural evolutionary process would reveal the existence of an attractor.

By introducing several novel evaluation methods, we estimate the extent to which successive transmission events affect the evolution of multiple text properties, namely its *toxicity*, *positivity*, *difficulty*, and *length*. By comparing the properties of the initial (human-generated) and final texts (after several transmissions between LLMs), we illustrate and study the existence of potential attractors in LLM cultural evolution. In particular, we measure the effect of consecutive interactions compared to the effect of the first one. To study how text properties and attractors are affected by the specific context in which culture evolves, we conduct our analyses on five different models (*ChatGPT-3.5-turbo-0125*, *Llama3-8B-Instruct*, *Mistral-7B-Instruct-v0.2*, *Llama3-70B-Instruct*, and *Mixtral-8x7B-Instruct-v0.1*), three different tasks (i.e., instructions to either "rephrase", "take inspiration from", or "continue" the initial text) and 20 different initial texts. Although our focus is on a better understanding of the cultural dynamics of LLMs, the metrics and evaluation methods introduced here may also be of interest to researchers studying human cultural evolution.

The code for reproducing the simulations, analyses and figures is available on our GitHub[2]

**Our main contributions are:**

- We propose that there might be a gap in current LLM evaluations methods (single-turn evaluations might not be suited to assess the properties of multi-turn interactions)
- We empirically confirm this hypothesis by showing that multi-turn interactions indeed often lead to distributions of text properties that are significantly different from what is observed after a single interaction (Section 4.2)
- We introduce novel conceptual and methodological tools to fill this gap, grounded in research in cultural evolution, and in particular the concept of cultural attractor (Section 3.4).
- We showcase the potential of this method by applying it to compare the effect of different tasks, of different models, of temperature, and of fine-tuning on the properties of multi-turn interactions. (Section 4.3).
- We find several robust effects, such as the fact that less constrained tasks lead to stronger attractors, that some properties posses stronger attractors than others, and that fine-tuning can shift the position and modify the strength of attractors.

## 2 RELATED WORK

**Biases in LLMs outputs** LLM-generated content is known to exhibit a variety of stereotypical biases (Bender et al., 2021; Weidinger et al., 2021; Bai et al., 2024). In single-turn settings, LLMs perpetrate (Nadeem et al., 2020) or even amplify human biases based on gender, nationality, race, and religion (Kotek et al., 2023). For instance, the GPT model was shown to exhibit cultural values similar to those of *WEIRD* (Western, Educated, Industrial, Rich, Democratic) cultures (Atari et al., 2023). LLMs trained through reinforcement learning with human feedback (RLHF) were found to overly express left-wing opinions on American politics — a tendency that, once formed, is difficult to avoid even after steering the model toward different demographic groups (Santurkar et al., 2023). Finally, a large body of work compared cognitive biases of models to those of humans (Coda-Forno et al. (2024); Liu et al. (2024); Binz and Schulz (2023)).

**Transmission chains featuring artificial agents** Several studies have applied experimental designs used in cultural evolution to study knowledge and skill accumulation in groups of Reinforcement

---

[2]https://github.com/jeremyperez2/TelephoneGameLLM

Learning agents (Cook et al., 2024; Schmitt et al., 2018; Team et al., 2021; Prystawski et al., 2023). Closer to the current study, populations of LLMs have also been studied (Brinkmann et al., 2023). Iterative chains of generative models trained on the preceding model's output have been shown to sometimes collapse toward the most likely outputs while the tails of the original distribution disappear (Shumailov et al., 2024; Peterson, 2024; Gerstgrasser et al., 2024; Kazdan et al., 2024; Dohmatob et al., 2024). This idea of using LLM-generated content to fine-tune the next generation has also been applied to groups of LLMs with various communication structures (Helm et al., 2024). Similar to our approach, iterative chains with frozen (i.e., not re-trained) LLMs have been shown to express human-like biases in terms of gender stereotypes, positivity, and social, threat, and biology-related information (Acerbi and Stubbersfield, 2023). Strong, but non-human-like biases for producing factual information have also been observed (Chuang et al., 2023), stressing the importance of understanding the evolution of content in LLMs and the ways it might deviate from human cultural evolution.

**Iterated Learning**    Parallel works ((Ren et al., 2024; Zhu and Griffiths, 2024)) have studied similar questions using different methodologies, namely the framework of Bayesian Iterated Learning (BIL) ((Kirby et al., 2014; Griffiths and Kalish, 2007)). Under this framework, attraction is conceptualized as convergence towards priors, whereas we conceptualize it as the fixed point of a recurrent sequence. Therefore, BIL allows to make prediction about how an agents' hypothesis about a given phenomenon will evolve through social interaction, while our method captures how a specific cultural trait will evolve. Moreover, our framework enables us to avoid assuming that LLMs possess "priors" in the same way that humans do, which might be problematic (see Kovač et al. (2023); Shanahan et al. (2023)).

**Evaluation of LLMs in social contexts**    While not focusing on the effect of multi-turn interactions, an important body of work has started to study the social cognition of LLMs. These studies revealed that GPT-4 struggles to reach human performance with respect to social commonsense reasoning and strategic communication skills ((Zhou et al., 2024)). In a similar vein, Huang et al. (2024) found that manipulating the communication structure can make LLM collectives more resilient to the introduction of malicious agents.

## 3 METHODS

### 3.1 EXPERIMENTAL PARADIGM: LLM TRANSMISSION CHAINS

In transmission chains, individual participants are ordered linearly. Each participant receives some information from the previous one, performs a task, and transmits new information to the next participant. Each agent is prompted with a **task** (instruction on how the text should be processed) and a **text**, which are concatenated and passed to the *user message*. The first agent is given a human-generated text and a task, and subsequent agents are given the same task and the text generated by the previous agent in the transmission chain: $text_{i+1} = LLM(task, text_i)$, where $text_0$ is the initial human-generated text and $LLM$ generates an output based on task $task$ and the previous agent's text $x_i$. We run this process for 50 generations. Examples of texts evolving through generations are provided in Appendix Section A.

**Initial texts ($text_0$)** We borrow human-generated text from various databases to provide the initial input to each transmission chain. Since we were interested in how variation in the initial text would impact the properties of the ensuing chain, human-generated texts spanned various types of content: scientific abstracts As we are interested in the evolution of the *toxicity*, *positivity*, *difficulty* and *length* of generated texts, we sample the entire dataset to obtain a subset of 20 initial texts that covered the range of possible values for these properties. The exact method used to extract these texts is detailed in Appendix A. As a robustness checks, we replicated our experiments with a larger set of 100 initial texts (Appendix B).

**Tasks** To determine the effects of instructions on the evolution of content over generations of LLMs, we prompt each chain of LLMs with three different tasks encompassing typical uses of LLMs:

*Rephrase*: agents are instructed to paraphrase the received text without modifying its meaning. This task is relevant for applications such as text simplification, or for content summarization.

*Take inspiration*: agents are instructed to take inspiration from the received text to produce a new one. It can be used in creative writing, where the goal is to generate new and original content.

*Continue*: agents are instructed to continue the received text. It is relevant for applications such as dialogue generation, in order to generate coherent and relevant responses to user inputs, or for content generation in storytelling and gaming.

Tasks remained consistent within each chain. The exact prompt used for each task is reported in Appendix Section A.

**Models** To assess whether and how cultural evolution dynamics are affected by the model specifications, we run identical experiments using six different models, all commonly used, from three different companies and with varying sizes: *GPT-4o-mini*, *GPT-3.5-turbo-0125* (referred to as "GPT3.5") , *Llama3-8B-Instruct* ("*Llama3-8B*"), *Mistral-7B-Instruct-v0.2* ("*Mistral-7B*"), *Llama3-70B-Instruct* ("*Llama3-70B*"), *Mixtral-8x7B-Instruct-v0.1* ("*Mixtral-8x7B*"). For inference, we used the OpenAI API (The MIT License) [3] to run *GPT-4o-mini* and *GPT3.5* and the HuggingFace's Transformer library (Wolf et al., 2019) for other models (Apache Licence, v2.0).

## 3.2 METRICS

Iterated transmissions may affect the generated text in several ways. We focus on four, orthogonal properties for each text which could be automatically measured:

*Toxicity*. Companies typically fine-tune LLMs to avoid the generation of toxic (i.e., dangerous or harmful) content. However, to our knowledge, this fine-tuning step focuses on single-turn dynamics, and the evolution of content with respect to its toxicity is currently understudied. We measure the toxicity of texts by quantifying the presence of rude, disrespectful, or unreasonable language, using a probability score that ranges from 0.0 (benign and non-toxic) to 1.0 (highly likely to be toxic), as estimated by the classifier introduced in Hanu and Unitary team (2020).

*Positivity*. Even when trained to avoid toxic content, LLMs have been shown to express similar negativity biases to humans, often favoring negative over positive information in preserving and generating new information (Acerbi and Stubbersfield, 2023). To study whether positivity biases over transmission chains are affected by tasks and models, we measure the positivity of produced contents using the `SentimentIntensityAnalyzer` tool from NLTK (Hardeniya et al., 2016). It uses this information to calculate a sentiment score for the text, ranging from -1.0 (highly negative) to 1.0 (highly positive).

*Difficulty*. While LLMs are argued to benefit society by democratizing knowledge (Weiss), such positive outcomes are conditioned on the LLMs generating output that is accessible and inclusive to all kinds of audiences. However, whether text difficulty is preserved, increased, or reduced over transmission chains is currently unknown. We estimate text difficulty using the Gunning-Fog index (Bogert, 1985), which depends on the average sentence length and the percentage of difficult words. A standard interpretation of this index is that it estimates the years of formal education required to fully understand the text.

*Length*: A simple, yet crucial aspect of a piece of text is its length. As more and more content is generated by and from LLM outputs, cultural media may become populated with increasingly short (potentially incomplete) or long (potentially redundant, meaningless, or hard to process) material. We therefore assess the evolution of content length as measured by the character count.

We provide additional details about metrics in Appendix Section A.

## 3.3 EFFECT OF MULTI-TURN TRANSMISSIONS

One of our questions is how content evolves over multi-turn transmissions compared to single-turn settings. To address this point, we compare the distribution of a given property in the generated texts at the first generation to the distributions at subsequent generations. Thus for each model and task, we look at the properties of each of the 100 generated texts (20 transmissions chains * 5 seeds) at each generation, which gives us a sample of 100 property values for each value. Using a Kolmogorov–Smirnov test (Massey, 1951), we then test whether the sample obtain at each generation

---

[3]https://openai.com/index/openai-api/

comes from the same distribution as the sample obtained after the first generation. If we can confidently reject the hypothesis that the sample of property values at the end of the transmission chain comes from the same distribution as the sample obtained after the first generation, this would confirm that looking at outputs after a single-turn transmission is not enough for predicting output properties in a multi-turn setting.

## 3.4 ATTRACTOR STRENGTH AND POSITION

Human cultural evolution shows that cultural traits sometimes evolve towards attractor states, i.e., content that invites convergence even with different starting points (Kalish et al., 2007; Miton et al., 2015; 2020; Buskell, 2017). Therefore, we were interested in whether transmission chains with LLMs would show similar attractor dynamics, and whether these depend on the model and task used in the chain. The concept of *cultural attractor* is not consistently formalized in the human cultural evolution literature (Buskell, 2017). Here, we defined attractors as the theoretical equilibrium point to which the iterated generation process (defined in 3.1) may eventually converge. We mathematically define attractors in terms of two properties of interest: its position (i.e., the location in output generation space the process converges toward) and its strength (i.e., the intensity to which generated outputs are pulled toward it). The strength takes values in $[0, 1]$, which allows for a continuous notion of an attractor: rather than being a binary concept that either exists or does not, attractors here lie on a spectrum, covering systems without attraction effects (strength=0) to ideal attractors (strength=1). To compute position and strength, we use the simulated data to fit a linear regression predicting the value of a property at the end of the chain (i.e. after $n_{generations}$ as a function of its value in the initial text (Figure S1).

For example for a given text *property*, we fit: $property(generation = n_{generations}) = I + s * property_{initial}$, where $I$ is the estimated intercept and $s$ the estimated slope. This enables us to estimate the final output of a new chain starting from the final output of the previous chain as: $property(generation = 2 * n_{generations}) = I + s * property(generation = n_{generations})$.

The fitted linear regression thus allows to define a recurrent relationship between the output of a chain as a function of the output of the previous chain: $property(generation = k * n_{generations}) = I + s * property(generation = (k-1) * 50)$. This relationship is a linear recurrence sequence which can be rewritten as: $property(generation = k * n_{generations}) = s^k * (property_{initial} - l) + l$, where $l = \frac{I}{1-s}$. If $|s| < 1$, then the sequence converges, its limit is $l$ and its convergence rate is $1 - s$. We can therefore use the estimated relationship to determine if an attractor exists ($|s| < 1$) and, if so, estimate its position $l = \frac{I}{1-s}$ and strength $1 - s$.

To validate that these theoretical fixed points correctly capture attraction dynamics, we estimated their positions using only data from the *first 10 generations* of each chain, and compared the predictions with the actual output after *50 generations*. Visual inspection of the results (Appendix B.5) confirmed that our method is suited for estimating the strength and position of attractors.

## 4 RESULTS

For each of the 6 models, 3 tasks, and 20 initial texts, we ran 5 transmission chains with 50 transmission steps. In the main experiment, each chain is composed of a population of agents sharing the same underlying model. In an additional experiment, we also studied chains where different models interact with one another (Appendix B.2.4). We provide some examples of generated texts in Appendix Section 1, and complete data on the companion website [4]. By extracting the properties of generated texts at each generation of each chain, we can study the evolution of these properties through generations, measure how they are affected by interactions beyond single-turn effects, as well as detect and characterize theoretical attractors.

## 4.1 QUALITATIVE ANALYSIS OF PROPERTY EVOLUTIONS OVER GENERATIONS

In Figure 2, we show the evolution of property distributions over generation for each model and task. This reveals important difference depending on the analyzed property, the task and the model. For

---

[4]https://sites.google.com/view/telephone-game-llm

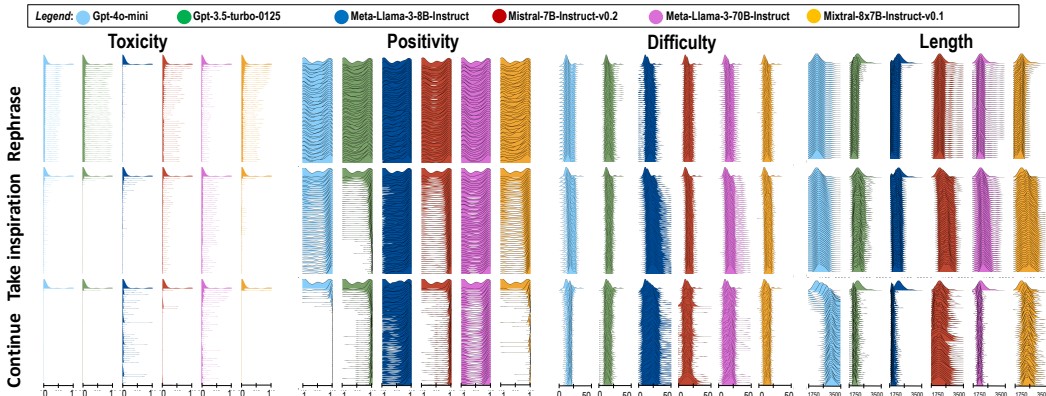

Figure 2: **Evolution of the distribution of text properties across generations.** We here represent the distribution of each of the four properties at each generation, for each model and task. These distributions thus represent the properties observed in the set of 100 transmission chains (20 initial texts * 5 seeds) for each model and task. For each property, task and model, the 50 generations are arranged vertically, with first generations at the top and last generations at the bottom.

instance, we observe that *toxicity* converges very quickly to a very narrow peak centered around 0. This is very different from the evolution of *positivity*, for which the initial distribution appears to be quite preserved for the *Rephrase* task (Figure 2, first row), while less constrained tasks such as *Take inspiration* (second row) and *Continue* (third row) lead to more visible changes. Interestingly, we observe that for *Llama3-8B* and *Llama3-70B*, the distribution of positivity values converges to a bimodal distribution, while distributions are unimodal for other models. In some cases, we also observe that different models lead the distributions to be shifted in opposite directions. For instance when looking at the evolution of texts *length*, using *GPT3.5* or *Llama3-8B* leads text to become on average shorter, while using *Mixtral-8x7B* or *GPT-4o-mini* shifts the distribution towards greater lengths.

## 4.2 To what extent do multi-turn transmissions affect the evolution of properties?

Qualitative analyses from the previous section appear to suggest that multi-turn transmissions lead texts to acquire different properties compared to single-turn settings. To quantitatively evaluate this observation, we use Kolmogorov–Smirnov (KS) tests (Massey, 1951) to estimate the compare property distributions after a single interaction and after multiple interactions (see Section B.3.1 for justification of why this method is adequate). In Figure 3, we report for each model, task and property the p-value of the KS test for the null hypothesis $H_0$: *"The text properties at generation $i$ are sampled from the same distribution as the text properties after generation 1"*. Across most instances, we observe that the p-values steadily decrease, indicating that observing the given distribution under the null hypothesis becomes less and less likely with generations. We observe that this is more often the case for less constrained tasks (*Take Inspiration* and *Continue*, second and third columns) than for more constrained task (*Rephrase*, first column). This finding confirms that studying single-turn interactions is in general not sufficient for analyzing the properties of interacting LLMs' outputs. This warrants a more detailed account of the cultural dynamics across iterated interactions among LLMs.

## 4.3 What influences the presence, strength, and position of attractors?

**Effect of model, task and property** Visual inspection of the evolution of text properties as presented on Figure 2 indicate that multi-turn transmissions lead distributions to become skewed toward certain values, which suggests the presence of attractors. The task assigned to a chain and the model type populating it appear to influence the position of those attractors, as well as their strength (i.e. how quickly do shifts in distributions happen). To have quantitative measures of attractors strengths and positions, we use the method described in 3.4 and Figure S1. Figure 4 presents the estimated strengths and positions of attractors, and fitted linear regressions are provided as supplementary material in

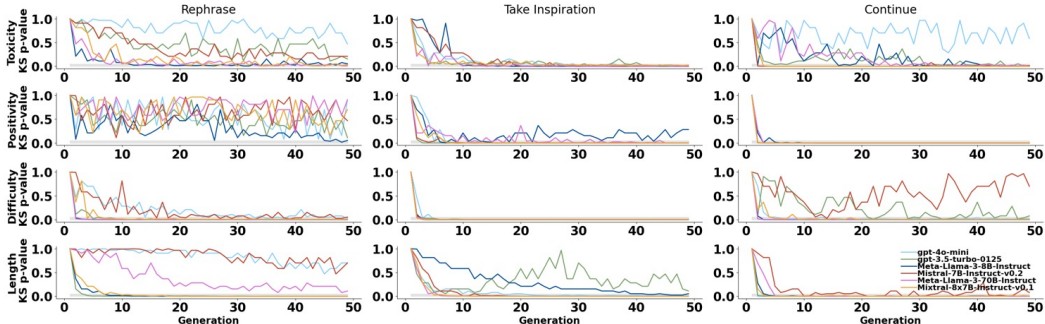

Figure 3: **Text properties are affected by transmissions beyond the first one.** p-values of the KS-test for the null hypothesis $H_0$: *"The text properties at generation $i$ are sampled from the same distribution as the text properties after generation 1"*, for each task (columns), property (rows) and models (colors). The grey shaded area represents p-values lower than 0.05. Over most instances, p-values decrease over generation and become close to 0, indicating that multi-turn transmissions lead to significantly different distributions compared to single-turn interactions.

Figure S2. For all combinations of property, task and model, we found that the recurrent relationship defined by the fitted linear regression converges. This means that all conditions admit a theoretical attractor as defined in Section 3.4. To better disentangle the respective contributions of model type, task and property on attractors position and strength, we fitted bayesian models predicting attractor **Strength** as a function of **Task**, **Model** and **Property**, and predicting attractor **Position** as a function of **Task** and **Model**, for each of the four considered properties. We use 95% Credible Intervals (CI) to assess significance. Those confidence intervals, as well as details of statistical models, are provided in Appendix section B.

We find a strong effect of **Task** on attractor strength: *Continue* leads to significantly stronger attraction than *Take Inspiration* , itself leading to significantly stronger attraction than *Rephrase* . This supports our observation that less constrained tasks lead to stronger attraction than more constrained tasks. To specifically test this hypothesis, we conducted additional experiments manipulating only the room for variation allowed in the task, which confirmed this hypothesis (Figure S5).

Different properties are also found to display different sensitivity to attraction effects. We detect that *toxicity* possesses significantly stronger attractors than *positivity*, *difficulty* and *length*. As for the effect of model, we observe significantly weaker attraction for *Llama3-70b* compared to *GPT3.5*, *Llama3-8B* and *Mixtral-8x7b*. *GPT-4o-mini* also displayed significantly weaker attractors than *GPT3.5*, *Llama3-8B* and *Mixtral-8x7b*.

As for the position of the attractors, we found that the position of the attractor for *positivity* was significantly lower for *Llama3-8b* than for *GPT3.5*, *GPT-4o-mini*, *Mistral-7b* and *Mixtral-8x7b*, and that the task *Take inspiration* and *Continue* both led to significantly higher *positivity* than the *Rephrase* task.

**Robustness checks** To verify the robustness of our results, we verified that they hold when using a larger set of initial texts (Figure S3) and when using different phrasings of the instructions prompts (Figure S4). We observed similar trends and values as in the main experiment, which confirms the robustness of our results.

**Effect of temperature** Increasing the temperature appears to lead to stronger attractors only in constrained tasks (*Rephrase* and *Take inspiration*), but not in the more open- ended task *Continue* (Figure 5a). One interpretation might be that increasing temperature relaxes constraints on the content that can be produced, thus leading to stronger attractors. This effect would therefore be more significant for tasks that are quite constrained.

**Effect of fine-tuning** We observe that fine-tuning can shift the attractors' positions (Figure 5b): the attractor for *length* is lower, and the attractor for *difficulty* higher, for Instruct models compared to Base models. For *toxicity* and *positivity*, we do not observe very significant shifts. This may indicate that Base models were already quite aligned with human preferences in terms of *toxicity* and *positivity*

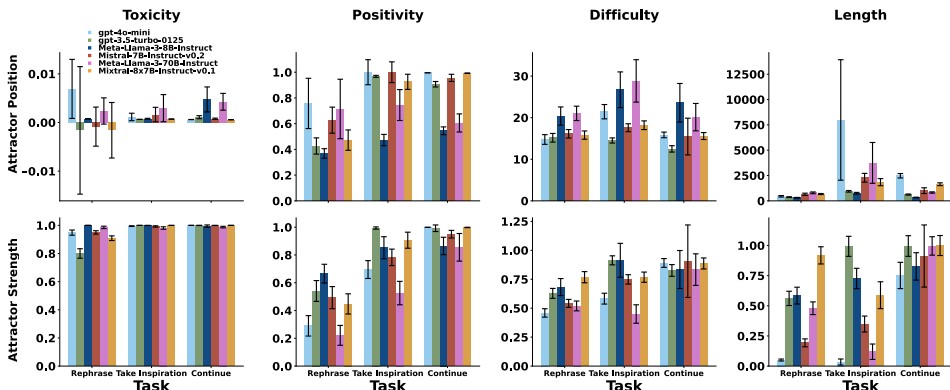

Figure 4: **Attractors strength and position.** The heigth of the bars represent the position (top row) and strength (bottom row) of theoretical attractors estimated using the method described in Section 3.4, for each property (columns), task, and model. Less constrained tasks, such as *Continue*, appear to produce stronger attractors than more constrained tasks, such as *Rephrase*. Attractors appear to be stronger for *toxicity* than for *length*. Finally, we can notice that the position of attractors appears to vary between models.

even before fine-tuning, possibly due to techniques such as data curation. *Length* and *difficulty* may have been less targeted by such techniques, and fine-tuning may therefore have shifted the attractors' positions towards human preferences. As for the strength of attractors, we observe that fine-tuning seems to increase attraction strength for *toxicity*, but to reduce it for *difficulty*. This may suggest that for properties on which humans have strong, uniform preferences (such as *toxicity*), fine-tuning leads to stronger attractors, while for properties on which humans have weaker and more heterogeneous preferences, it mitigates the strength of attractors that came from the training data. This analysis provides insight about how attractors are formed, and reveals that both the choice of training data and fine-tuning processes may impact LLMs in ways that only become significant in the case of multi-turn behaviors.

## 5 DISCUSSION

While current studies analyzing the outputs of LLMs are restricted to a single prompt-output interaction, we borrowed the methodology from studies on human cultural evolution to address how cultural content may evolve over transmission chains with LLMs. This resulted in a series of *telephone game* experiments assessing the evolution of cultural content in LLMs as a function of models, instructions, and text properties. Our results reveal that several changes in generated content appear after multiple iterations. For example, we observed that the *difficulty* of a provided text was preserved after an LLM was prompted to elaborate it a single time, but changed dramatically after the text was processed iteratively by a chain of LLMs. Those qualitative observations were confirmed by statistical tests, where we found that multi-turn interactions lead the distributions of text properties to become significantly different distributions obtained after single-turn interactions.

By comparing the properties of input texts to those of texts produced by transmission chains spanning several generations of LLMs, we identified property-specific patterns in the convergence of LLM dynamics toward attractor states. Using this method, we found that some properties (e.g. *toxicity*) display quicker convergence rates toward attractors, and that the position of these attractors varies between models. Moreover, our results reveal that more open-ended tasks (e.g. *Continue*) lead to quicker convergence toward attractors than more constrained tasks (e.g. *Rephrase*). Studying attraction dynamics in systems of LLMs may therefore be particularly relevant to situations in which LLMs are used to simulate artificial societies (de Zarzà et al., 2023; Park et al., 2023; 2022; Xiao et al., 2023; Hua et al., 2023; Vezhnevets et al., 2023; Chuang et al., 2023), where they are often granted a relatively high freedom. Evaluating the effect of temperature on attractors supports this view that allowing more room for variation leads to stronger attractors, as we found that increasing temperature increases attraction strength only for quite constrained tasks. To understand the role

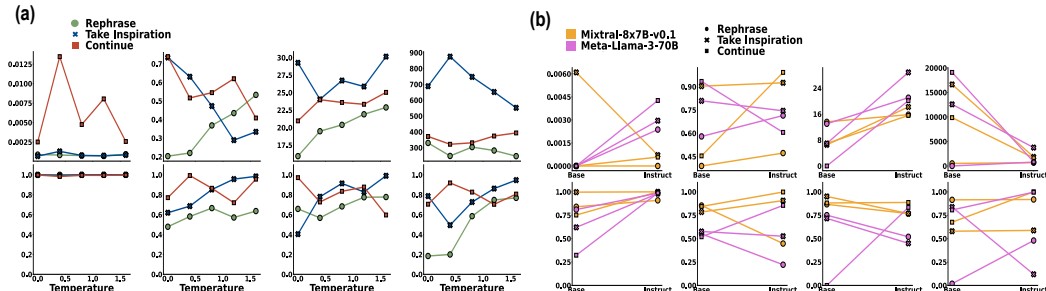

Figure 5: **Effect of temperature (a) and fine-tuning (b) on attractors. (a)** Attractor positions (top row) and strength (bottow row) for different values on temperature (x-axis), for model *Llama3-8B-Instruct*. The main visible effect is that increasing temperature increases attraction strength for tasks *Rephrase* and *Take Inspiration*, but not for *Continue*. **(b)** Attractor positions (top row) and strength (bottow row) for *Base* and *Instruct* versions of *Mixtral-8x7B* and *Llama3-70B*. Fine-tuning appears to increase the strength of attraction for *toxicity*, increases the position of the attractor for *difficulty*, and decreases the position of the attractor for *length*.

of fine-tuning in shaping the strength and position of these attractors, we compared *Base* models with their *Instruct* versions. This revealed that fine-tuning can affect both attractors strength (e.g. for *toxicity*) and position (e.g. for *difficulty* and *length*).

The main practical consequence of our results is that they question the current ways in which LLMs are fine-tuned and evaluated. Indeed, current fine-tuning procedures, such as RLHF, rely on a single interaction to rate the output of the LLMs. Similarly, the vast majority of benchmarks and leaderboards are based on the capacities of LLMs in single-turn interactions. As multi-step interaction is a major use-case of LLMs (e.g. when interacting iteratively with a chatbot, or when LLMs are used in multi-agent simulations), there is a contradiction between how LLMs are used and how LLMs are evaluated and trained. Our results indicate that this contradiction cannot be ignored: for example, on Figure S26 (*toxicity - take inspiration*) we see that at first Mistral appears less toxic at the first iteration, but after 9 generations *Llama3-8B* ends up being much less toxic. One may therefore choose *Mistral-7B* based on single-step evaluation, even though *Llama3-8B* is the optimal choice for many applications.

More generally, our results reveal that knowing the output of LLMs after a single interaction is insufficient to predict their behavior in long-term social interactions. Therefore, this suggests that existing training and evaluation methods are not suited to align the behavior of LLMs in multi-step settings. The concepts of attractor position and strength may offer a way to take into account the consequence of multi-turn interactions when comparing and evaluating LLMs.

**Limitations and future work** As our results indicate that both training data and fine-tuning affect attractors properties, future work should further explore how different fine-tuning and data curation methods shape multi-turn dynamics. While we focused on linear transmission chains, real-world interactions typically involve networks of senders and receivers. Human studies have shown that network size (Fay et al., 2019; Richerson, 2013; Andersson and Read, 2014; Baldini, 2015; Derex et al., 2013) and structure (Raviv et al., 2020; Kirby and Tamariz, 2022; Derex and Boyd, 2016; de Pablo et al., 2022; Derex and Mesoudi, 2020) influence cultural evolution. Following some initial endeavors (Nisioti et al., 2022; Perez et al., 2024; Lai et al., 2024), future work may assess similar effects in machine networks, and in particular investigate how attraction dynamics are shaped by network-based strategies such as dynamic network structures (Nisioti et al., 2022) or free-formed decentralized networks (Lai et al., 2024). To investigate model- and task-specific biases, we focused on homogeneous transmission chains where agents belonged to the same model type and received the same instructions. Future studies may address cultural dynamics in heterogeneous populations of LLMs prompted with diverse instructions. Our study could also be extended to hybrid networks in which humans and LLMs interact — a scenario that is becoming increasingly relevant as generative tools become more widespread and which may, in turn, shape the future of human cultural evolution (Brinkmann et al., 2023).

## REPRODUCIBILITY STATEMENT

The code for reproducing simulations, statistical analyses, and for generating all figures can be found on our GitHub repository: https://github.com/jeremyperez2/TelephoneGameLLM. This repository also contains the data generated when running the experiments. This data can also be accessed from the companion website (https://sites.google.com/view/telephone-game-llm), using the Data Explorer tool.

## ETHICS STATEMENT

The results presented in this paper highlight that multi-turn interactions can give rise to potential harmful biases that cannot be detected by only studying the outcome of single-turn interactions. By introducing novel conceptual and methodological tools, we provide ways of quantifying the properties of multi-turn interaction systems with respect to the extent to which they magnify or reduce these biases. It is then the responsibility of end-users to make an ethical use of these tools, for instance by using them to ensure that LLMs remain aligned with ethical standards even after repeated interactions.

## ACKNOWLEDGEMENTS

This research was partially funded by the French National Research Agency (ANR, project ECOCURL, Grant ANR-20-CE23-0006). This work benefited from access to the Jean Zay (Idris) supercomputer associated with the Genci grant A0151011996. We also thank Chris Foulon, Marcela Ovando-Tellez and Joan Dussauld who participated in the hackathon Hack1Robo during which this project originated. M.D. acknowledges Institute for Advanced Study in Toulouse (IAST) funding from the French National Research Agency (ANR) under grant no. ANR-17-EURE-0010 (Investissements d'Avenir programme).

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

## A   ADDITIONAL DETAILS ON THE METHODS

**Selecting initial texts**   We extracted 5 scientific abstracts [5], 10 news articles [6], and 5 social media comments [7] from online datasets as initial texts. To ensure that those initial texts covered the range of text properties we were interested in, we proceeded as follows: for *difficulty*, we measured the maximal and minimal *difficulty* $d_{min}$ and $d_{max}$ of texts from the scientific abstracts datasets, defined a linear space of 5 values $(d_i)_{i=1:5}$ between $d_{min}$ and $d_{max}$ and sampled 5 texts, each having a value of difficulty close to $(d_i)_{i=1:5}$. We then followed the same procedure for *toxicity*, using the dataset of social media comments; for *positivity*, using the dataset of news articles; *length*, using the dataset of news articles.

**Pre-processing outputs**   Data analyses revealed that, on the *Continue* task, when using Mistral-7B, agents of the chains would sometimes start outputting very long text by filling them with "*#some_keyword*". As this behavior created a few outliers, we thought it would be better to filter-out those "*#some_keyword*" when performing the main analyses. This behavior is nevertheless an interesting result, reminiscent of the collapsing dynamics found when training LLMs on their own output (Shumailov et al., 2023). We therefore discuss it separately in Appendix B.

**Hyperparameters**   We use the following hyperparameters for generations in all models. Temperature was set to $0.8$ with and top_p to $0.95$. All models, except GPT3.5, bfloat16 precision was used.

**Computational resources**   Experiments were conducted with the OpenAI API (less than 5 million tokens), and with a cluster equipped with A100, and V100 GPU graphic cards. Running the final experiments for the four models, which were run on the cluster, required less than 3000 GPU hours. Experiments with *Llama3-8B* and *Mistral-7B* were conducted on V100 NVIDIA GPUs with 32GB of VRAM, and experiments with *Llama3-70B* and *Mixtral-8x7B* on two A100 NVIDIA GPUs with 80GB VRAM in parallel.

**Environmental footprint**   Taking heat recovery in account, running experiments consumed about 777kWh, which represents about 15.54 kg of $CO_2$.

**Prompts used**   In our experiments, each task was induced by a specific instruction (prompt), which is given to each agent in the chain. For the *Rephrase* task, the instruction is: "You will receive a text. Your task is to rephrase this text without modifying its meaning. Just output your new text, nothing else. Here is the text:", for the *Inspiration* task, the instruction is: "You will receive a text. Your task is to create a new original text by taking inspiration from this text. Just output your new text, nothing else. Here is the text:", and for the *Continue* task, the instruction is: "You will receive a text. Your task is to continue this text. Just output your new text, nothing else. Here is the text:".

**Examples of stories**   Here we provide examples of stories that were given as input and stories that were generated in the last iteration of some chains. Table 1 shows one example for each task. Complete data can be found on the companion website [8] using the Data Explorer tool.

**Measuring text properties**

- *Toxicity*. We assess the level of toxicity in generated texts using the Detoxify library, a classifier developed for the Jigsaw Toxic Comment Classification Challenges (see `https://github.com/unitaryai/detoxify/tree/master` . This classifier defines toxicity as the presence of rude, disrespectful, or unreasonable language in a text and assigns a probability score ranging from 0.0 (benign and non-toxic) to 1.0 (highly likely to be toxic). Trained on a large dataset of human-labeled comments from various online platforms, the

---

[5]https://huggingface.co/datasets/CCRss/arxiv_papers_cs
[6]https://huggingface.co/datasets/RealTimeData/bbc_latest
[7]https://huggingface.co/datasets/FredZhang7/toxi-text-3M/blob/e0e5b168b4a7e14e84f07271bfe1c6b42bc91ccd/multilingual-train-deduplicated.csv
[8]https://sites.google.com/view/telephone-game-llm

classifier use a transformer-based architecture to analyze the text's context and meaning, identifying patterns indicative of toxicity.

- *Positivity.* We employ the `SentimentIntensityAnalyzer` tool from the NLTK library to assess the positivity of generated texts. The tool is based on the Valence Aware Dictionary and sEntiment Reasoner (VADER) method (Hutto and Gilbert, 2014), which is a lexicon and rule-based sentiment analysis tool specifically designed for social media data. It uses a combination of lexical features, such as words and their semantic orientation, to determine the overall sentiment of a text. In the VADER method, every word in the vocabulary is rated with respect to its positive or negative sentiment and the intensity of that sentiment. The `SentimentIntensityAnalyzer` uses this information to calculate a sentiment score for the text, ranging from -1.0 (highly negative) to 1.0 (highly positive).

- *Difficulty.* We estimate the difficulty of generated texts using the Gunning-Fog index. In the method, complex words are defined as those with three or more syllables, excluding proper nouns, familiar jargon (and removing common suffixes as syllables). We used textstat to compute it.

Table 1: Examples of starting stories and the corresponding final stories for chains with GPT3.5 on the three tasks

| Description | Story |
| --- | --- |
| Starting story | You are certainly no friend of mine. You have accused me of lying about what you had said, and run away when I proved by actual quotation that the accusation was false. We both know that, even though you praise humility, you are not humble enough to apologize. You sneered at my combat experience, saying "A lot of so called "combat veterans" of Vietnam smelled more perfume from call girls than they smelled cordite" and "So when you were an Army officer in Vietnam (even though you spent your time inside of Danang airbase, with at least two battalions of infantry marines on the wire) what was the combat load for M-16 ammo per soldier on a daily patrol?" An actual friend would not have written such things. No, you are no friend of mine, you ignorant, arrogant, hypocritical blow-hard. You should be able to guess that I do not feel friendly towards you |
| Final story (Continue) | Your laughter and light will always be a beacon of hope in our darkest days. Your presence may be gone, but your spirit will forever be alive in our hearts. We will carry on your legacy of kindness and love, spreading your positivity to all those we encounter. Your memory will never fade, as we hold you close in our thoughts and actions. Rest easy, knowing that you are surrounded by our love and gratitude. You may be gone, but you will never be forgotten. Until we meet again, dear friend, know that you are always with us. |
| Final story (Inspiration) | Amidst the chaos of our busy lives, it is essential to pause and prioritize forming genuine connections with those around us. By being fully present and showing empathy, we can cultivate relationships that are deep and fulfilling. Authenticity is the key to creating bonds that bring true joy and contentment into our lives. Engaging in open and sincere dialogues, sharing our vulnerabilities, and striving to forge meaningful connections are all essential in nurturing authentic relationships. Let's come together to create a community where empathy and mutual respect are the guiding principles, and where every interaction is infused with love and gratitude. These authentic connections serve as a support system rooted in compassion and generosity, uplifting and inspiring each other as we navigate the ups and downs of life. Together, we can weave a network of relationships that exude positivity and light, enriching our lives in meaningful and transformative ways. |
| Final story (Rephrase) | I need to end our friendship since you won't apologize for accusing me of dishonesty, ignoring evidence, and making disrespectful comments about my time in Vietnam. Your behavior shows a lack of empathy, arrogance, and insincerity, qualities that are not in line with those of a true friend. |

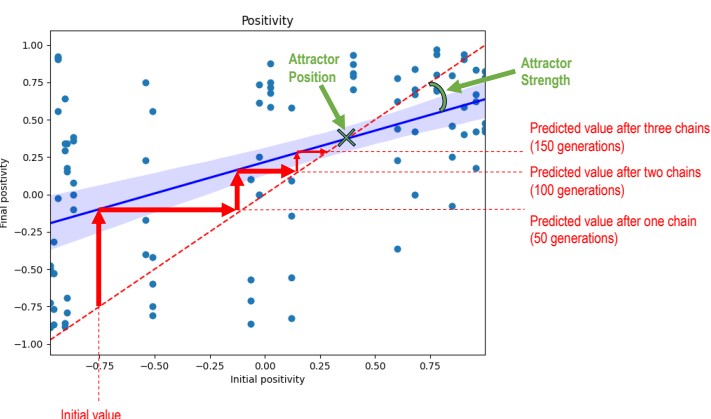

Figure S1: **Method for estimating attractor strength and position** This figures depicts the method introduced in Section 3.4 to estimate the strength and position of theoretical attractors. Each dot in this figure corresponds to one chain, for a total of 100 chains (20 initial texts * 5 seeds). The position of a dot on the x-axis corresponds to the value of the property (*positivity* in this example) in the initial text, while the position on the y-axis corresponds to the value of this property of the text produced after 50 generations. We then used these 100 data points to fit a linear regression predicting the relationship between the initial and final values of the property.

# B   ADDITIONAL FIGURES AND ANALYSES

## B.1   FITTED LINEAR REGRESSIONS

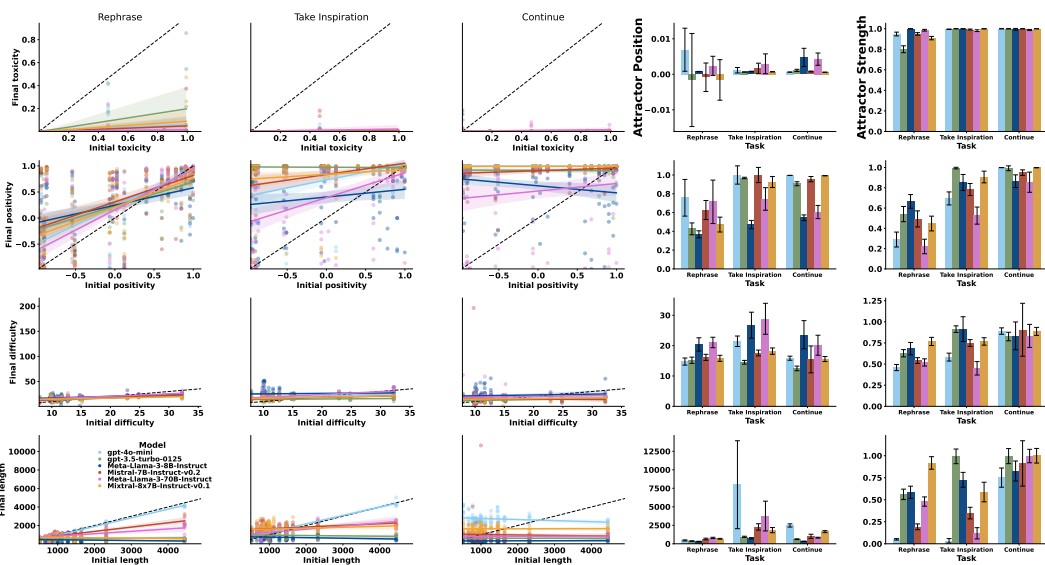

Figure S2: **Fitted linear regressions used to compute attractors strength and position.** For three tasks (Rephrase, Take Inspiration, Continue), five models, and three metrics (*toxicity*, *positivity*, *difficulty*, *length*), we plot (Mean ± SE) the relationship between the metric value of the initial human-written text (input to the first agent) and the value of the final LLM-generated text (output of the last agent). A slope close to zero indicates strong attraction, while the value at the intersection with the diagonal captures the position of the attractor.

## B.2 ROBUSTNESS CHECK AND CONTROLS

### B.2.1 INCREASING THE SIZE OF THE SET OF INITIAL TEXTS

To verify that the results presented in the main text hold when using a larger set of initial texts, we ran additional experiments with Meta-Llama-3-8b using 100 different initial texts, using the same sampling procedure as for the main experiment (see Section A). We observe that the trends remain the same with 100 initial texts as with 20 initial texts, and that the standard deviation intervals for 100 initial texts include – or are very close to including – the values estimated with 20 initial texts (with only one exception out of 24 values (Figure S3)

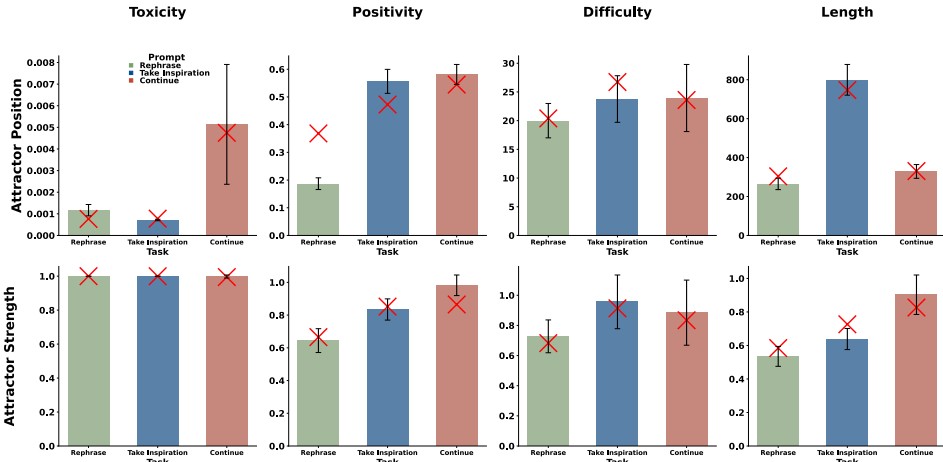

Figure S3: **Attractor strenghts and positions over 100 initial texts** Attractor position (first row) and strength (second row) for four text properties (column) and three tasks (colors). The height of the bars are the position and strength of the attractors for a set of 100 initial texts. The red crosses indicate the values estimated in the main experiment using a set of 20 initial texts (Section 4.3). This reveals that our results are robust to increasing the sample size of the initial texts dataset.

### B.2.2 DIFFERENT PHRASING OF THE INSTRUCTIONS PROMPTS

To verify that the results presented in the main text hold when using different phrasings of the initial prompts, we ran additional experiments with *Llama-3-8B* using 5 different phrasings of instructions prompts for each of the three. Those exact prompts are as follows:

**Rephrase**

- "You will be given a text. Your job is to reword this text without changing its meaning. Only provide your revised text, nothing else. Here is the text:"
- "A text will be provided to you. Your task is to rephrase it, keeping its meaning intact. Only output the rephrased text, and nothing additional. Here is the text:"
- "You'll receive a text, and your job is to rephrase it without altering its meaning. Just output your new version, nothing more. Here is the text:"
- "A text will be sent to you. Your role is to rephrase it while keeping the meaning the same. Only display your new text, with nothing extra. Here is the text:"
- "You will be provided with a text. Your task is to reword it without changing the intended meaning. Output just your rephrased text, nothing else. Here is the text:"

**Inspiration**

- "You'll be given a text, and your task is to craft a new, original text inspired by it. Only provide your new version, with nothing additional. Here is the text:"
- "A text will be provided to you. Your task is to create an original text based on this inspiration. Just output your new version, nothing extra. Here is the text:"
- "You will receive a text. Your job is to generate an original text inspired by it. Only show your new version, without adding anything else. Here is the text:"
- "You'll receive a text, and your task is to create a new text inspired by it. Simply display your new version, with no additional output. Here is the text:"
- "You will be sent a text. Your role is to produce an original text inspired by it. Only present your new text, nothing more. Here is the text:"

**Continue**

- "You'll be provided with a text. Your job is to extend this text. Only output your continuation, nothing additional. Here is the text:"
- "A text will be given to you. Your task is to carry on from this text. Just display your new continuation, with nothing extra. Here is the text:"
- "You'll receive a text. Your role is to continue from where this text ends. Only present your extension, with no additional output. Here is the text:"
- "A text will be sent to you, and your job is to complete it by continuing from its end. Only output your continuation, nothing more. Here is the text:"
- "You will get a text. Your task is to extend it further. Just display your continuation, without adding anything else. Here is the text:"

This revealed that our results are robust to different paraphrasing of the same prompt: indeed, the interval of the standard deviations over the 5 paraphrased prompts always contains – or almost contains – the values estimated with the original prompt in the main experiment. (Figure S4)

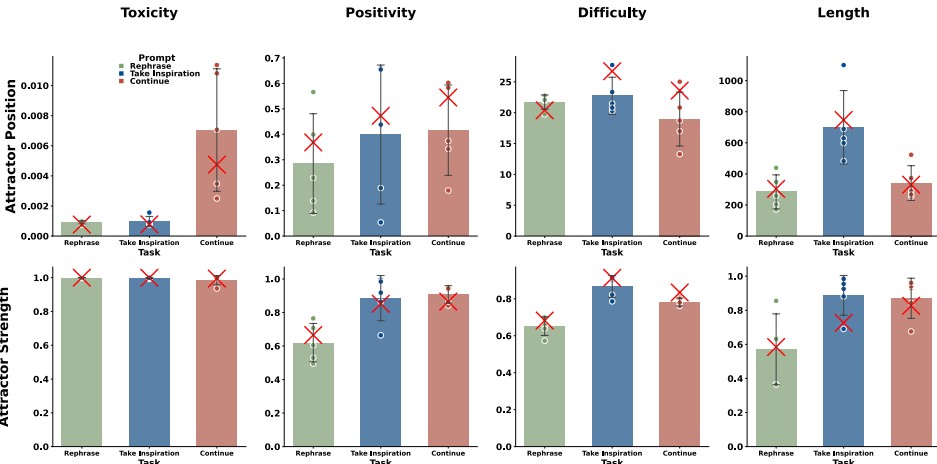

Figure S4: **Attractor strenghts and positions for different phrasings of the intruction prompts**
Attractor position (first row) and strength (second row) for four text properties (column) and three
tasks (colors). The height of the bars represent the average position and strength of the attractors over
the five different phrasings. Individual dots correspond to each different phrasing. The red crosses
indicate the values estimated in the main experiment using a set of 20 initial texts (Section 4.3). This
reveals that our results are robust to increasing the sample size of the initial texts dataset.

### B.2.3 EFFECT OF LESS VERSUS MORE CONSTRAINED TASKS ON ATTRACTORS

In the results from the main experiments, we observed that less constrained tasks (such as *Continue*) lead to stronger attractors than more constrained tasks (such as *Rephrase*). However, these two tasks vary on other dimensions than only the room for variation allowed. To verify our hypothesis, we conducted additional experiments with *Llama-3-8B* where we instructed the LLM to create a new text by making either "very minor", "a few small", "moderate", "notable" or "radical" changes to the received text.

The exact prompts were as follows:

- "You will receive a text. Your task is to create a new original text by taking inspiration from this text, making only very minor changes. Just output your new text, nothing else. Here is the text:"

- "You will receive a text. Your task is to create a new original text by taking inspiration from this text, making a few small changes. Just output your new text, nothing else. Here is the text:"

- "You will receive a text. Your task is to create a new original text by taking inspiration from this text, making moderate changes. Just output your new text, nothing else. Here is the text:"

- "You will receive a text. Your task is to create a new original text by taking inspiration from this text, making notable changes. Just output your new text, nothing else. Here is the text:"

- "You will receive a text. Your task is to create a new original text by taking inspiration from this text, making radical changes. Just output your new text, nothing else. Here is the text:"

This experiment confirmed our initial interpretation, as less constrained tasks indeed exhibit stronger attractors (Figure S5). The only exception is Length. We interpret this by the fact that going from "very minor" to "radical" changes mainly refers to the semantic of the text. As a consequence, the instructions to make "minor changes" is not more constrained than to make "radical changes" with respect to Length.

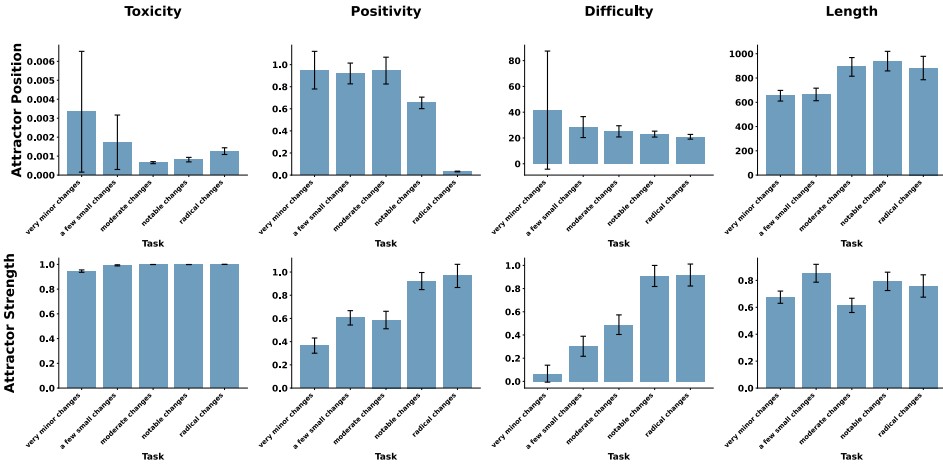

Figure S5: **Effect of less versus more constrained tasks on attractors** Attractor position (first row) and strength (second row) for four text properties (column) as a function of the room for variation allowed in the instruction. These results confirm our hypothesis that less constrained tasks lead to stronger attractors. The only exception is Length. We interpret this by the fact that going from "very minor" to "radical" changes mainly refers to the semantic of the text. As a consequence, the instructions to make "minor changes" is not more constrained than to make "radical changes" with respect to Length.

### B.2.4 EFFECT OF HETEROGENEOUS TRANSMISSION CHAINS

As the study of LLMs' multi-turn dynamics remains unexplored, it appeared necessary to start with a setting where causal variables can be isolated, so as to lay the foundation for incrementally improving our understanding of LLMs cultural dynamics. As a consequence, the main experiments focused on homogeneous chains where the multi-turn dynamics of different models can be assessed in isolation. However, we also investigate the effect of having heterogeneous chains, where two different models interact (Figure S6). We compared 4 conditions: homogeneous chains of Mistral-7B models, homogeneous chains of Llama-3-8B models, heterogeneous chains of Mistral-7B and Llama-3-8B models starting with Mistral-7B and heterogeneous chains of Mistral-7B and Llama-8B models starting with Llama-3-8B. We found that although the position of the attractor in heterogeneous chains is often in between the position of the homogeneous chains' attractors, it is not systematically the case.

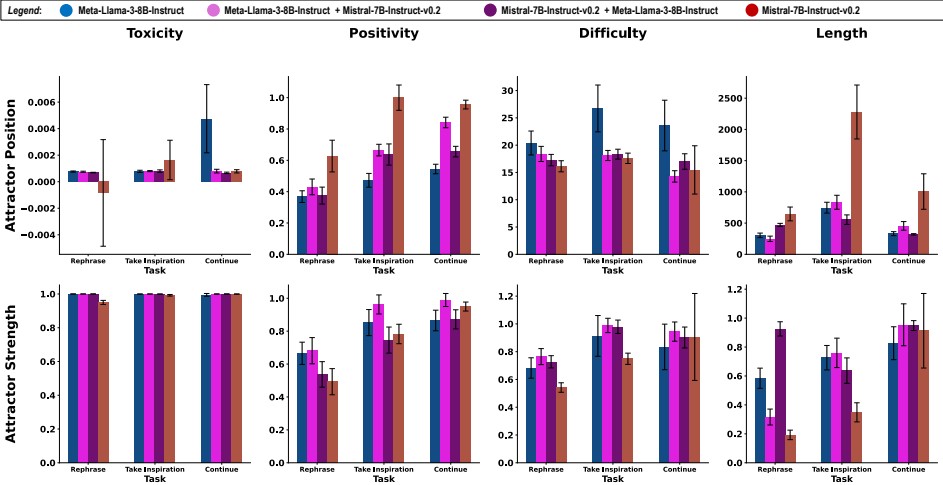

Figure S6: **Comparison of homogeneous and heterogeneous transmission chains** Attractor position (first row) and strength (second row) for four text properties (column) and three different tasks, for homogeneous chains of Llama-3-8B models (blue), homogeneous chains of Mistral-7B models (red), heterogeneous chains of Mistral-7B and Llama-3-8B models starting with Mistral-7B (dark purple) and heterogeneous chains of Mistral-7B and Llama-8B models starting with Llama-3-8B (light purple).

## B.3 STATISTICAL MODELS

### B.3.1 KOLMOGOROV-SMIRNOV TESTS

To quantitatively evaluate whether multi-turn transmission lead to significantly different outcomes than single-turn transmission, we use Kolmogorov–Smirnov (KS) tests (Massey, 1951) to estimate the compare distributions of text properties after a single interaction and after multiple interactions. Although we calculate p-values repeatedly (once for each generation, model and task), in our cases applying corrections to control for the False Discovery Rate (FDR) was in our case not necessary. Indeed, controlling for the FDR using methods such as Bonferroni correction is necessary in cases where many statistical tests are conducted, as it increases the likelihood that at least one of them is found significant "by chance" (Family-Wise Error Rate; Colas, 2022). With respect to our set-up, this would have been necessary if we concluded that multi-turn interactions lead to significant differences when at least one of the 49 p-values (one by generation) was below the significance threshold. However, we rather report all p-values, and draw conclusions from the trend observed over all generations, rather than from single p-values. Therefore, if we observe a general trend of decreasing p-values, we can conclude that observing the given distribution under the null hypothesis becomes less and less likely with generations. FDR corrections are thus unnecessary for drawing this conclusion.

### B.3.2 BAYESIAN MODELS

We performed statistical analyses using the Python package *pymc* (Wiecki et al., 2024) to fit Bayesian models.

- **Model 1** We fitted a model predicting the attractor strength (Figure 4) as a function of the Task, Model and Property: $Strength \sim \mathcal{N}(\mu, \sigma^2)$

  where $\mu = \alpha_{Task} + \beta_{Model} + \gamma_{Property}$

  Priors for parameters $a$, $b$ and $c$ were standard normal distribution, and standard half-normal distribution for $\sigma$.

- **Model 2** For each Property, we fitted a model predicting the attractor position (Figure 4) as a function of the Task and Model: $Position_{property} \sim \mathcal{N}(\mu, \sigma^2)$

  where $\mu = \alpha_{Task} + \beta_{Model}$

To determine the significance of the difference between estimated parameters, we computed the 95% credibility intervals of the difference by sampling from the posteriors. In Tables 4 to 12, we provide those credibility intervals. Intervals that do not contain 0 signal statistically significant differences.

|  | rephrase | inspiration | continue |  |
|---|---|---|---|---|
| rephrase | [0.0000 ; 0.0000] | [-0.2372 ; -0.0353] | [-0.4187 ; -0.2195] |  |
| inspiration | [0.0353 ; 0.2372] | [0.0000 ; 0.0000] | [-0.2824 ; -0.0840] |  |
| continue | [0.2195 ; 0.4187] | [0.0840 ; 0.2824] | [0.0000 ; 0.0000] |  |

Table 2: 95% Credible Intervals for posterior differences between prompts for attractor strength

|  | gpt-4o-mini | gpt-3.5-turbo-0125 | Meta-Llama-3-8B-Instruct | Mistral-7B-Instruct-v0.2 | Meta-Llama-3-70B-Instruct | Mixtral-8x7B-Instruct-v0.1 |  |
|---|---|---|---|---|---|---|---|
| gpt-4o-mini | [0.0000 ; 0.0000] | [-0.3539 ; -0.0699] | [-0.3289 ; -0.0451] | [-0.2361 ; 0.0487] | [-0.1644 ; 0.1208] | [-0.3501 ; -0.0649] |  |
| gpt-3.5-turbo-0125 | [0.0699 ; 0.3539] | [0.0000 ; 0.0000] | [-0.1161 ; 0.1673] | [-0.0228 ; 0.2603] | [0.0490 ; 0.3346] | [-0.1385 ; 0.1478] |  |
| Meta-Llama-3-8B-Instruct | [0.0451 ; 0.3289] | [-0.1673 ; 0.1161] | [0.0000 ; 0.0000] | [-0.0500 ; 0.2351] | [0.0256 ; 0.3087] | [-0.1617 ; 0.1209] |  |
| Mistral-7B-Instruct-v0.2 | [-0.0487 ; 0.2361] | [-0.2603 ; 0.0228] | [-0.2351 ; 0.0500] | [0.0000 ; 0.0000] | [-0.0686 ; 0.2134] | [-0.2561 ; 0.0277] |  |
| Meta-Llama-3-70B-Instruct | [-0.1208 ; 0.1644] | [-0.3346 ; -0.0490] | [-0.3087 ; -0.0256] | [-0.2134 ; 0.0686] | [0.0000 ; 0.0000] | [-0.3282 ; -0.0430] |  |
| Mixtral-8x7B-Instruct-v0.1 | [0.0649 ; 0.3501] | [-0.1478 ; 0.1385] | [-0.1209 ; 0.1617] | [-0.0277 ; 0.2561] | [0.0430 ; 0.3282] | [0.0000 ; 0.0000] |  |

Table 3: 95% Credible Intervals for posterior differences between model for attractor strength

| | toxicity | positivity | difficulty | length | |
|---|---|---|---|---|---|
| toxicity | [0.0000 ; 0.0000] | [0.1345 ; 0.3626] | [0.1274 ; 0.3584] | [0.2422 ; 0.4747] | |
| positivity | [-0.3626 ; -0.1345] | [0.0000 ; 0.0000] | [-0.1201 ; 0.1094] | [-0.0054 ; 0.2273] | |
| difficulty | [-0.3584 ; -0.1274] | [-0.1094 ; 0.1201] | [0.0000 ; 0.0000] | [-0.0014 ; 0.2330] | |
| length | [-0.4747 ; -0.2422] | [-0.2273 ; 0.0054] | [-0.2330 ; 0.0014] | [0.0000 ; 0.0000] | |

Table 4: 95% Credible Intervals for posterior differences between measures for attractor strength

| | gpt-4o-mini | gpt-3.5-turbo-0125 | Meta-Llama-3-8B-Instruct | Mistral-7B-Instruct-v0.2 | Meta-Llama-3-70B-Instruct | Mixtral-8x7B-Instruct-v0.1 | |
|---|---|---|---|---|---|---|---|
| gpt-4o-mini | [0.0000 ; 0.0000] | [-0.0012 ; 0.0069] | [-0.0033 ; 0.0049] | [-0.0016 ; 0.0064] | [-0.0043 ; 0.0037] | [-0.0010 ; 0.0070] | |
| gpt-3.5-turbo-0125 | [-0.0069 ; 0.0012] | [0.0000 ; 0.0000] | [-0.0061 ; 0.0020] | [-0.0045 ; 0.0035] | [-0.0071 ; 0.0009] | [-0.0039 ; 0.0042] | |
| Meta-Llama-3-8B-Instruct | [-0.0049 ; 0.0033] | [-0.0020 ; 0.0061] | [0.0000 ; 0.0000] | [-0.0024 ; 0.0056] | [-0.0051 ; 0.0030] | [-0.0019 ; 0.0063] | |
| Mistral-7B-Instruct-v0.2 | [-0.0064 ; 0.0016] | [-0.0035 ; 0.0045] | [-0.0056 ; 0.0024] | [0.0000 ; 0.0000] | [-0.0066 ; 0.0014] | [-0.0034 ; 0.0046] | |
| Meta-Llama-3-70B-Instruct | [-0.0037 ; 0.0043] | [-0.0009 ; 0.0071] | [-0.0030 ; 0.0051] | [-0.0014 ; 0.0066] | [0.0000 ; 0.0000] | [-0.0007 ; 0.0073] | |
| Mixtral-8x7B-Instruct-v0.1 | [-0.0070 ; 0.0010] | [-0.0042 ; 0.0039] | [-0.0063 ; 0.0019] | [-0.0046 ; 0.0034] | [-0.0073 ; 0.0007] | [0.0000 ; 0.0000] | |

Table 5: 95% Credible Intervals for posterior differences between model for attractor position - toxicity

| | rephrase | inspiration | continue | |
|---|---|---|---|---|
| rephrase | [0.0000 ; 0.0000] | [-0.0031 ; 0.0000] | [-0.0040 ; -0.0009] | |
| inspiration | [-0.0000 ; 0.0031] | [0.0000 ; 0.0000] | [-0.0025 ; 0.0006] | |
| continue | [0.0009 ; 0.0040] | [-0.0006 ; 0.0025] | [0.0000 ; 0.0000] | |

Table 6: 95% Credible Intervals for posterior differences between prompt for attractor position - toxicity

| | gpt-4o-mini | gpt-3.5-turbo-0125 | Meta-Llama-3-8B-Instruct | Mistral-7B-Instruct-v0.2 | Meta-Llama-3-70B-Instruct | Mixtral-8x7B-Instruct-v0.1 | |
|---|---|---|---|---|---|---|---|
| gpt-4o-mini | [0.0000 ; 0.0000] | [-0.0982 ; 0.3959] | [0.2023 ; 0.6959] | [-0.1887 ; 0.3048] | [-0.0172 ; 0.4735] | [-0.1299 ; 0.3673] | |
| gpt-3.5-turbo-0125 | [-0.3959 ; 0.0982] | [0.0000 ; 0.0000] | [0.0521 ; 0.5498] | [-0.3395 ; 0.1533] | [-0.1712 ; 0.3262] | [-0.2771 ; 0.2166] | |
| Meta-Llama-3-8B-Instruct | [-0.6959 ; -0.2023] | [-0.5498 ; -0.0521] | [0.0000 ; 0.0000] | [-0.6410 ; -0.1471] | [-0.4723 ; 0.0229] | [-0.5813 ; -0.0818] | |
| Mistral-7B-Instruct-v0.2 | [-0.3048 ; 0.1887] | [-0.1533 ; 0.3395] | [0.1471 ; 0.6410] | [0.0000 ; 0.0000] | [-0.0767 ; 0.4176] | [-0.1860 ; 0.3139] | |
| Meta-Llama-3-70B-Instruct | [-0.4735 ; 0.0172] | [-0.3262 ; 0.1712] | [-0.0229 ; 0.4723] | [-0.4176 ; 0.0767] | [0.0000 ; 0.0000] | [-0.3566 ; 0.1401] | |
| Mixtral-8x7B-Instruct-v0.1 | [-0.3673 ; 0.1299] | [-0.2166 ; 0.2771] | [0.0818 ; 0.5813] | [-0.3139 ; 0.1860] | [-0.1401 ; 0.3566] | [0.0000 ; 0.0000] | |

Table 7: 95% Credible Intervals for posterior differences between model for attractor position - positivity

| | rephrase | inspiration | continue | |
|---|---|---|---|---|
| rephrase | [0.0000 ; 0.0000] | [-0.4662 ; -0.1133] | [-0.4465 ; -0.0938] | |
| inspiration | [0.1133 ; 0.4662] | [0.0000 ; 0.0000] | [-0.1570 ; 0.1913] | |
| continue | [0.0938 ; 0.4465] | [-0.1913 ; 0.1570] | [0.0000 ; 0.0000] | |

Table 8: 95% Credible Intervals for posterior differences between prompt for attractor position - positivity

| | gpt-4o-mini | gpt-3.5-turbo-0125 | Meta-Llama-3-8B-Instruct | Mistral-7B-Instruct-v0.2 | Meta-Llama-3-70B-Instruct | Mixtral-8x7B-Instruct-v0.1 | |
|---|---|---|---|---|---|---|---|
| gpt-4o-mini | [0.0000 ; 0.0000] | [-2.6049 ; 2.8520] | [-2.9991 ; 2.4390] | [-2.6157 ; 2.7268] | [-2.9831 ; 2.4137] | [-2.6832 ; 2.7637] | |
| gpt-3.5-turbo-0125 | [-2.8520 ; 2.6049] | [0.0000 ; 0.0000] | [-3.1499 ; 2.2758] | [-2.7682 ; 2.6207] | [-3.1053 ; 2.2984] | [-2.8099 ; 2.6017] | |
| Meta-Llama-3-8B-Instruct | [-2.4390 ; 2.9991] | [-2.2758 ; 3.1499] | [0.0000 ; 0.0000] | [-2.3698 ; 3.0545] | [-2.7175 ; 2.7392] | [-2.4107 ; 3.0411] | |
| Mistral-7B-Instruct-v0.2 | [-2.7268 ; 2.6157] | [-2.6207 ; 2.7682] | [-3.0545 ; 2.3698] | [0.0000 ; 0.0000] | [-3.0124 ; 2.3806] | [-2.6899 ; 2.6762] | |
| Meta-Llama-3-70B-Instruct | [-2.4137 ; 2.9831] | [-2.2984 ; 3.1053] | [-2.7392 ; 2.7175] | [-2.3806 ; 3.0124] | [0.0000 ; 0.0000] | [-2.3859 ; 3.0178] | |
| Mixtral-8x7B-Instruct-v0.1 | [-2.7637 ; 2.6832] | [-2.6017 ; 2.8099] | [-3.0411 ; 2.4107] | [-2.6762 ; 2.6899] | [-3.0178 ; 2.3859] | [0.0000 ; 0.0000] | |

Table 9: 95% Credible Intervals for posterior differences between model for attractor position - difficulty

| | rephrase | inspiration | continue | |
|---|---|---|---|---|
| rephrase | [0.0000 ; 0.0000] | [-3.0072 ; 2.2777] | [-2.6556 ; 2.6528] | |
| inspiration | [-2.2777 ; 3.0072] | [0.0000 ; 0.0000] | [-2.2601 ; 2.9840] | |
| continue | [-2.6528 ; 2.6556] | [-2.9840 ; 2.2601] | [0.0000 ; 0.0000] | |

Table 10: 95% Credible Intervals for posterior differences between prompt for attractor position - difficulty

| | gpt-4o-mini | gpt-3.5-turbo-0125 | Meta-Llama-3-8B-Instruct | Mistral-7B-Instruct-v0.2 | Meta-Llama-3-70B-Instruct | Mixtral-8x7B-Instruct-v0.1 | |
|---|---|---|---|---|---|---|---|
| gpt-4o-mini | [0.0000 ; 0.0000] | [-1.8746 ; 3.6795] | [-1.8366 ; 3.7029] | [-2.1079 ; 3.4919] | [-2.2197 ; 3.3266] | [-2.0495 ; 3.4024] | |
| gpt-3.5-turbo-0125 | [-3.6795 ; 1.8746] | [0.0000 ; 0.0000] | [-2.7134 ; 2.8132] | [-2.9561 ; 2.5396] | [-3.1474 ; 2.4007] | [-2.9893 ; 2.5194] | |
| Meta-Llama-3-8B-Instruct | [-3.7029 ; 1.8366] | [-2.8132 ; 2.7134] | [0.0000 ; 0.0000] | [-3.0299 ; 2.5205] | [-3.1786 ; 2.3807] | [-3.0437 ; 2.4675] | |
| Mistral-7B-Instruct-v0.2 | [-3.4919 ; 2.1079] | [-2.5396 ; 2.9561] | [-2.5205 ; 3.0299] | [0.0000 ; 0.0000] | [-2.9116 ; 2.6355] | [-2.7884 ; 2.7556] | |
| Meta-Llama-3-70B-Instruct | [-3.3266 ; 2.2197] | [-2.4007 ; 3.1474] | [-2.3807 ; 3.1786] | [-2.6355 ; 2.9116] | [0.0000 ; 0.0000] | [-2.6617 ; 2.8778] | |
| Mixtral-8x7B-Instruct-v0.1 | [-3.4024 ; 2.0495] | [-2.5194 ; 2.9893] | [-2.4675 ; 3.0437] | [-2.7556 ; 2.7884] | [-2.8778 ; 2.6617] | [0.0000 ; 0.0000] | |

Table 11: 95% Credible Intervals for posterior differences between model for attractor position - length

| | rephrase | inspiration | continue | |
|---|---|---|---|---|
| rephrase | [0.0000 ; 0.0000] | [-4.1791 ; 1.3268] | [-3.1691 ; 2.4195] | |
| inspiration | [-1.3268 ; 4.1791] | [0.0000 ; 0.0000] | [-1.7409 ; 3.8265] | |
| continue | [-2.4195 ; 3.1691] | [-3.8265 ; 1.7409] | [0.0000 ; 0.0000] | |

Table 12: 95% Credible Intervals for posterior differences between prompt for attractor position - length

### B.4 DISCONTINUITIES AND COLLAPSING BEHAVIOR

In the main text, the experiment with the Mistral-7B model on the *Continue* task was analyzed by first filtering the hashtags, as discussed in Appendix A. Given, that this behavior is interesting in itself, we discuss it here in more details.

Figure S7 shows the average length of text generated with the Mistral-7B model chain on the *Continue* task for five different seeds of the same story. We can observe several discontinuities in terms of the generated text length, i.e. at some iterations the length drastically increases or decreases. It is interesting to note that when the length decreases, it returns to the original value as before the first discontinuity. This suggests the existence of an attractor regarding this specific length. To better understand the cause of these discontinuities, Figure 13 shows examples of stories generated before and after those discontinuities (for the seed number three in figure S7). We can see that at generation 14 the model abruptly starts to generate many hashtags. It generates 283 hashtags, compared to 12 in the previous generation. At generation 45, we can see that the overall quality of the text decreased into generating solely hashtags and brief descriptions. This reduction in text quality is reminiscent of collapsing dynamics observed in iterative chains of LLMs, where each model was trained on the output of a previous one (Shumailov et al., 2023).

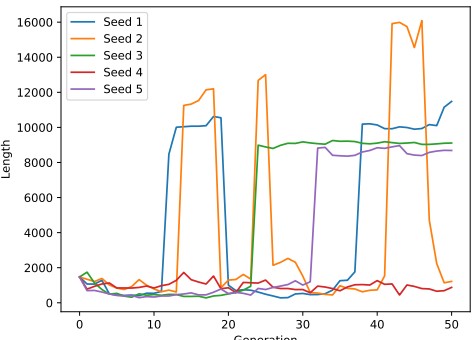

Figure S7: **Discontinuities and collapse in the Mistral-7B model chain** The lengths of generated stories are shown (without filtering out the hashtags) for five chains starting with the same initial story. We observe discontinuities, where the length drastically increases or decreases. After decreasing, the length of the story goes back to the original length, suggesting the existence of an attractor.

Table 13: Examples of stories demonstrating the collapse and recovery in Mistral-7B chains on the *Continue* task for (seed 2 from figure S7

| Gen. | Story |
|---|---|
| 13 | We are thrilled about the progress we've made in making Xiangqi more accessible for all. Let's keep the conversation going and explore new ways to adapt the game for different abilities. Share your ideas, experiences, and success stories using the hashtags #CXAInclusiveXiangqi #XiangqiForAll #AccessibleXiangqi #XiangqiUnited #DisabilityInclusion #AdaptiveXiangqi #InclusiveGaming #AccessibleCommunity. Let's continue to learn from each other and create a vibrant, inclusive Xiangqi community that celebrates diversity and welcomes everyone. #UnitedWeGame #AccessibleXiangqiJourney #TogetherWeCan #XiangqiForEveryone |
| 14 | Let's exchange innovative ideas on modifying Xiangqi pieces, boards, and rules to accommodate various disabilities. #AdaptiveXiangqiDesigns #InclusiveXiangqiSolutions #TogetherWeAdapt #XiangqiEmpowerment #AccessibleXiangqiProgress #DisabilityFriendlyXiangqi #XiangqiInclusiveCommunity #BreakingBarriers #XiangqiForAllPlayers #AccessibleXiangqiChampions #XiangqiInclusionSuccessStories #XiangqiUnitedForAll #AccessibleXiangqiFuture #InclusiveXiangqiVision
*(omitted 264 hashtags for clarity)*
#XiangqiInclusiveGamingCommunityVision #XiangqiAccessibleGamingCommunityGrowth #XiangqiAccessibleGamingCommunityInnovation #XiangqiAccessibleGamingCommunityEmpowerment #XiangqiAccessibleGamingCommunityPassion |
| 45 | #DesignWithInclusiveDesignPhilosophyScaling: Embracing diversity and equality in design practices.
#DesignWithUserCenteredDesignPhilosophyScaling: Putting users first in design decisions and experiences.
#DesignWithInclusiveTechnologyPhilosophyScaling: Making technology accessible to all users, regardless of abilities.
#DesignWithDigitalInclusionPhilosophyScaling: Ensuring everyone has equal access to digital resources and services.
*(omitted 176 lines for clarity).*
#DesignWithUserTestingTrainingScaling: Scaling user testing training opportunities.
#DesignWithAssistiveTechnologyTrainingScaling: Expanding assistive technology training opportunities.
#DesignWithInclusive |
| 49 | #DesignWithGlobalAccessibilityInitiativesScaling: Expanding global accessibility initiatives and collaborations.
*(omitted 7 lines for clarity)*
#DesignWithInclusiveDesignTrendsScaling: Growing trends and innovations in inclusive design and accessibility.
#DesignWithInclusiveDesignResourcesScaling: Expanding resources for inclusive design and accessibility knowledge and tools. |

## B.5 Validation of attractors position and strength estimation

The method introduced in Section 3.4 gives the position and strength of a theoretical attractor (or theoretical fixed point). In order to validate our method, we verified that this theoretical prediction matches the actual data. To do so, we used the first 10 generations of each simulated chain to predict the strength and position of attractors for each task, model and property. We then compared this prediction with the actual properties of texts obtained after 50 generations. As shown in Figure S8, transmission chains shifts the initial distribution of values in the direction of the predicted attractor. Moreover, the variance of the final distribution appears to reflect the predicted strength of the attractor. These results confirmed that the method we introduce is indeed suited for estimating the strength of position of attractors.

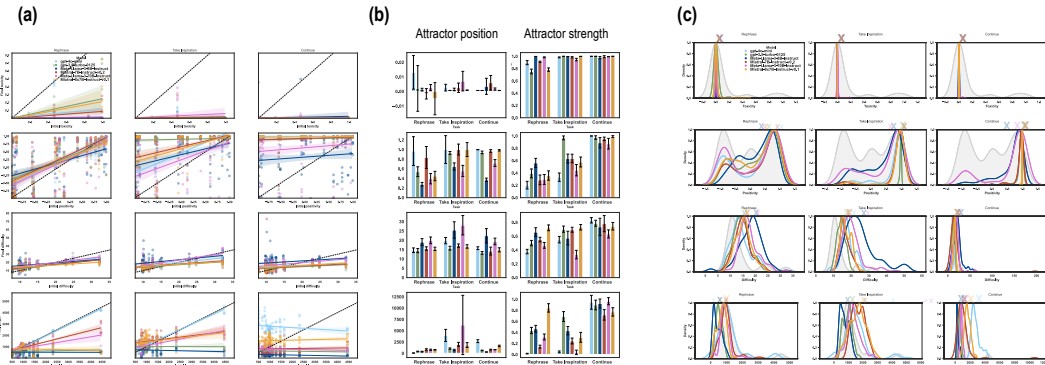

Figure S8: **Empirical validation of attractors position and strength estimation.** To empirical verify that the method introduced in Section 3.4 makes accurate prediction, we used the first 10 generations of each chain to fit the linear regression between initial and final property values **(a)**. We then used our method to estimate attractors' strength and position **(b)**. We then compared those predictions with the actual shifts in distribution observed after 50 generations **(c)**. The grey area represents the initial distribution of the corresponding property, and colored line show the distribution after 50 generations for each model. Crosses indicate the estimated position of theoretical attractors, and their size represent its strength. For the fourth row, second column, one attractor was outside the range of represented values and is thus represented with "-> X".

## B.6 EVOLUTION OF TEXT PROPERTIES FOR ALL INITIAL STORIES

We here provide the figures representing the evolution of each of the four metric for each model, for each of the 20 initial stories. Lines represented the average over 5 seeds, and shaded areas represent the standard errors.

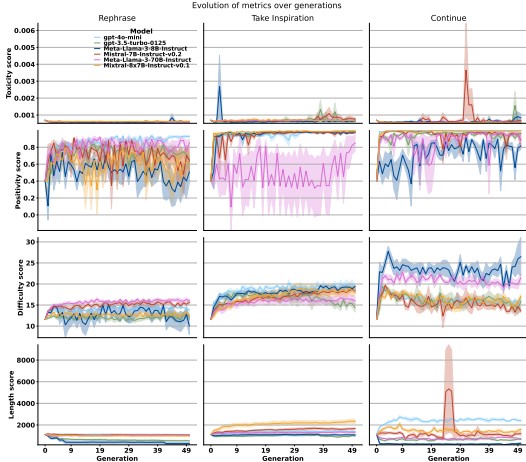

Figure S9: **Evolution of text properties starting with Initial Text 1**

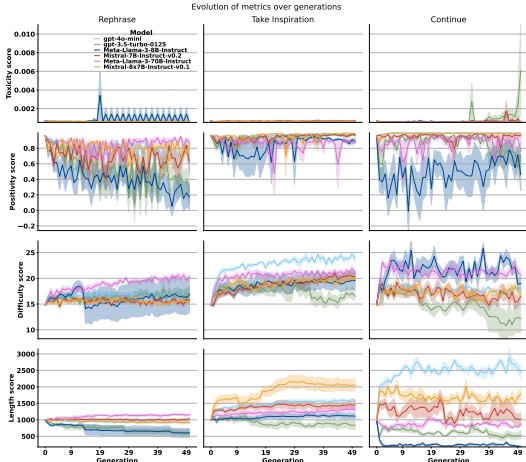

Figure S10: **Evolution of text properties starting with Initial Text 2**

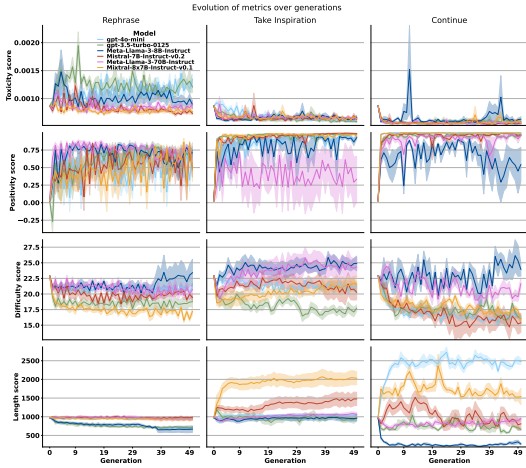

Figure S11: **Evolution of text properties starting with Initial Text 3**

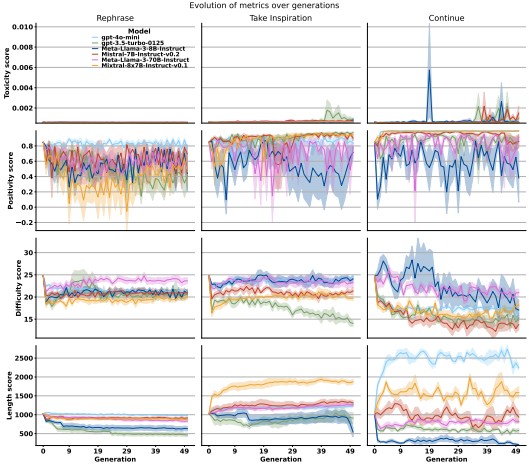

Figure S12: **Evolution of text properties starting with Initial Text 4**

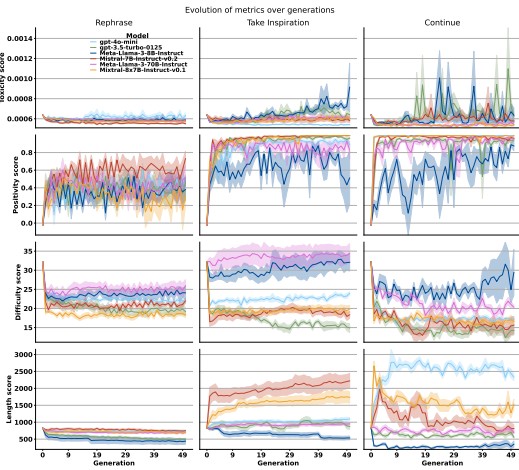

Figure S13: **Evolution of text properties starting with Initial Text 5**

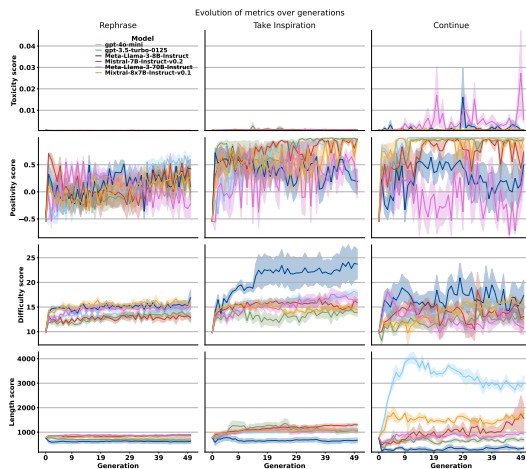

Figure S14: **Evolution of text properties starting with Initial Text 6**

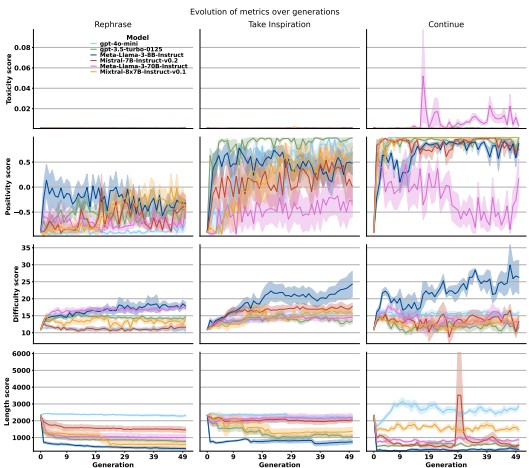

Figure S15: **Evolution of text properties starting with Initial Text 7**

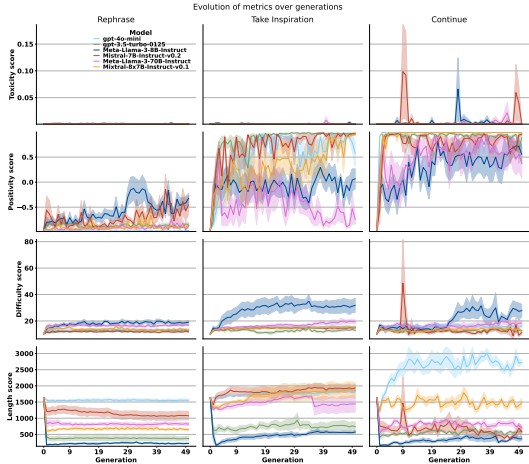

Figure S16: **Evolution of text properties starting with Initial Text 8**

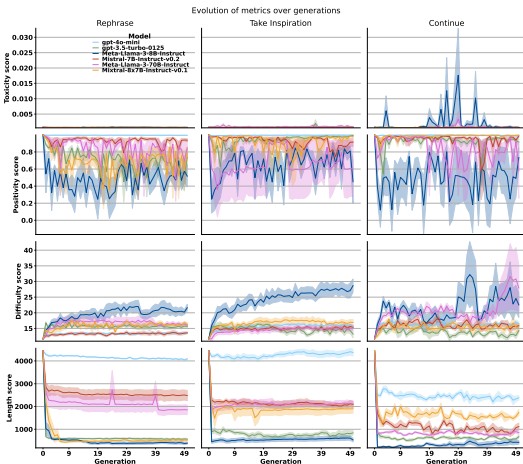

Figure S17: **Evolution of text properties starting with Initial Text 9**

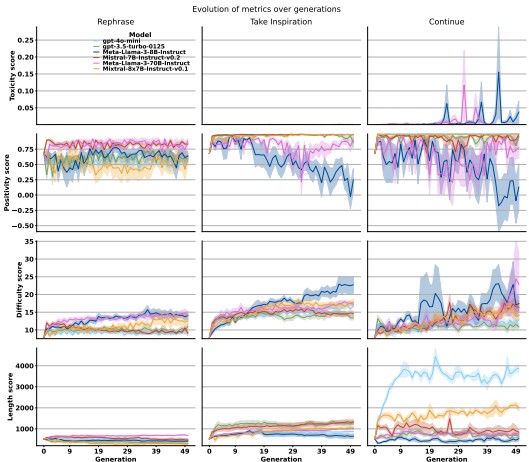

Figure S18: **Evolution of text properties starting with Initial Text 10**

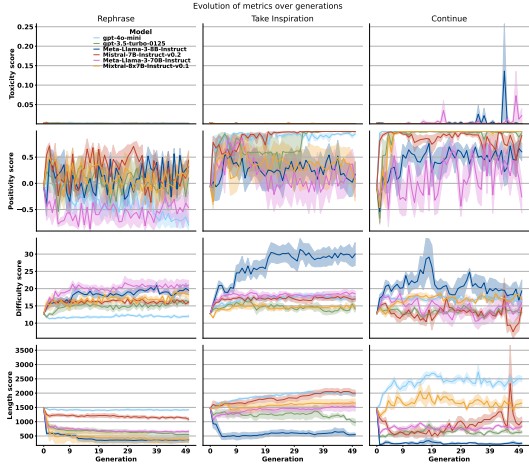

Figure S19: **Evolution of text properties starting with Initial Text 11**

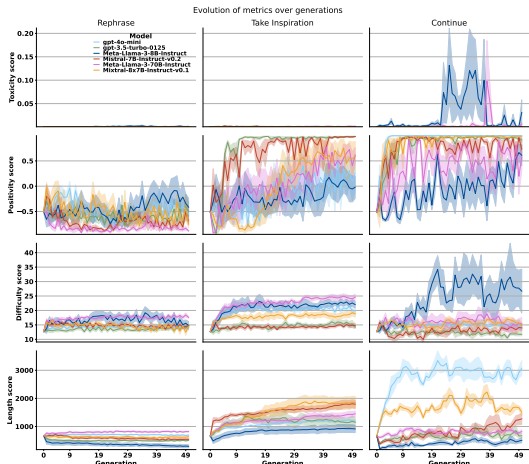

Figure S20: **Evolution of text properties starting with Initial Text 12**

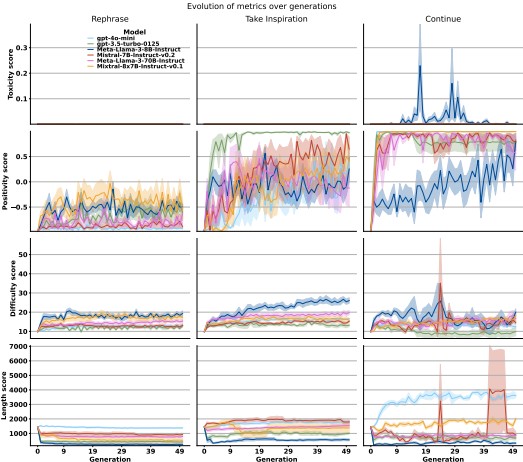

Figure S21: **Evolution of text properties starting with Initial Text 13**

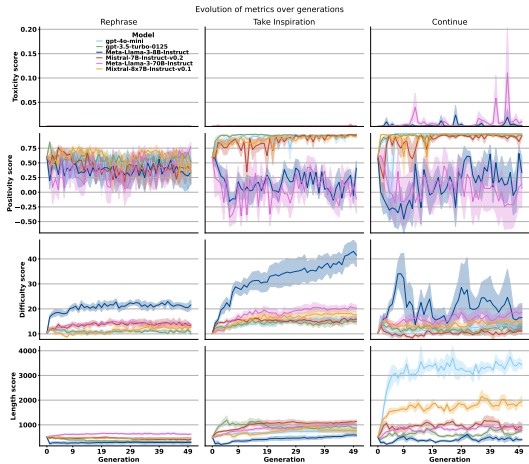

Figure S22: **Evolution of text properties starting with Initial Text 14**

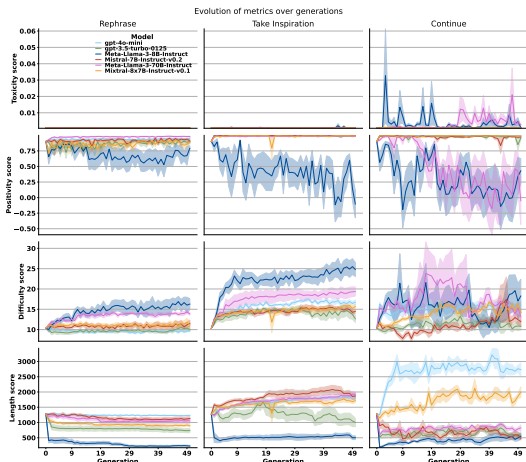

Figure S23: **Evolution of text properties starting with Initial Text 15**

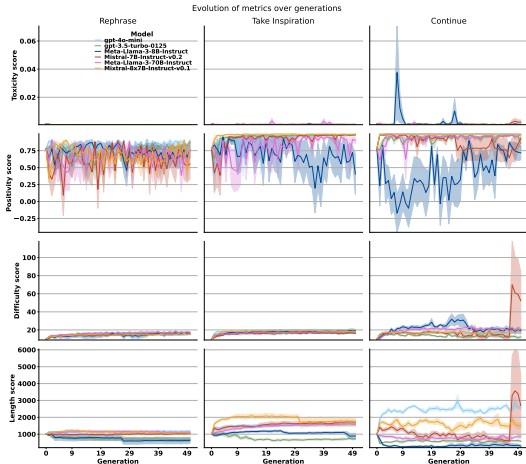

Figure S24: **Evolution of text properties starting with Initial Text 16**

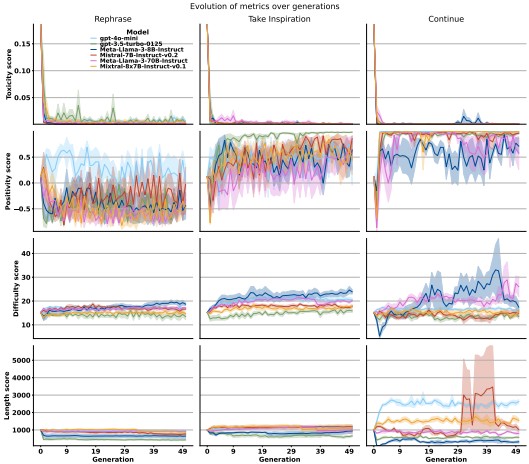

Figure S25: **Evolution of text properties starting with Initial Text 17**

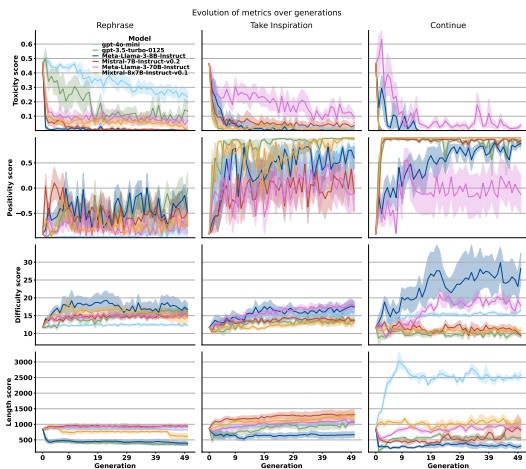

Figure S26: **Evolution of text properties starting with Initial Text 18**

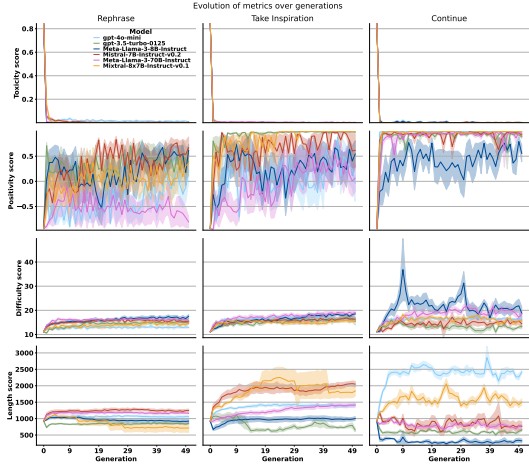

Figure S27: **Evolution of text properties starting with Initial Text 19**

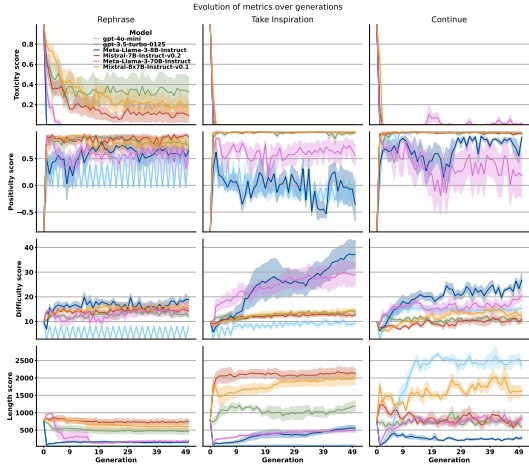

Figure S28: **Evolution of text properties starting with Initial Text 20**

