# OpenReview forum: "When LLMs Play the Telephone Game: Cultural Attractors as Conceptual Tools to Evaluate LLMs in Multi-turn Settings"
_ICLR.cc/2025/Conference — ICLR 2025 Poster_

### Official Review · Reviewer_uDks · 2024-11-03

**Soundness:** 2
**Presentation:** 2
**Contribution:** 2
**Rating:** 5
**Confidence:** 4

**Summary:**

The paper “When LLMs Play the Telephone Game” explores how repeated information transitions between language models (LLMs) change the properties of the information over time. In a series of experiments, the authors show that as text moves through multiple LLMs, small biases can build up, shifting content towards stable “attractor” states. They find that factors like model type, task instructions, and tuning settings affect how strongly text properties like toxicity or positivity converge. This study highlights that evaluating LLMs on single outputs misses important dynamics, calling for multi-turn testing to understand LLMs’ behavior in iterative information transitions.

**Strengths:**

The paper explores the shift of information properties during multi-turn LLM interactions using a “telephone game” setup. The experiment design is systematic, with some important metrics like toxicity and positivity, effectively showing how text properties shift across iterations. The paper contains a complex analysis, introducing some new methods for evaluating cultural attractors. Its findings are significant, emphasizing that single-turn evaluations miss key dynamics, especially for applications involving iterative or multi-agent LLM use. This work offers insights for improving LLM evaluation and alignment in real-world scenarios.

**Weaknesses:**

Although I appreciate the authors' attempt at quantifying the "loss" of information during multi-turn agent interactions, a major concern is that the setup and tasks (rephrase, continue, and inspire) are not diverse or realistic enough to reflect the real-world use case or issues that people may face for social simulation. For example, the work only explores linear chains, missing out on complex network structures that mimic real-world LLM interactions (as indicated in the limitations). The findings obtained from the work highlight the importance of understanding/evaluating LLMs dynamically have been explored in many other previous works.
E.g., [SOTOPIA: Interactive Evaluation for Social Intelligence in Language Agents](https://arxiv.org/abs/2310.11667), [On the Resilience of Multi-Agent Systems with Malicious Agents](https://arxiv.org/abs/2408.00989). I found the paper insightful and learned a few things from it, though I feel it may benefit from further development to fully meet the standards for a conference publication.

**Questions:**

For different kinds of tasks, have you tried different formats of the prompts? I would imagine the prompt would change quite a lot for some findings in the paper.

---

> ### Author Response · Authors · 2024-11-20
>
> *Due to the character limit on OpenReview, our response will be posted in two parts.*
>
> **Part 1 / 2:**
>
> We thank the reviewer for their thorough review and insightful comments. We are encouraged by the fact that the reviewer found the paper insightful, the findings significant, and the implications of our study useful for evaluating and aligning LLMs in real-world scenarios.
>
> We address each of their comments below:
>
>
>
> > *a major concern is that the setup and tasks (rephrase, continue, and inspire) are not diverse or realistic enough to reflect the real-world use case or issues that people may face for social simulation. For example, the work only explores linear chains, missing out on complex network structures that mimic real-world LLM interactions (as indicated in the limitations).*
>
> We thank the reviewer for inviting us to comment on the ecological validity of our experimental design. The real-life dynamics of LLM content generation are indeed much more complex than our experimental setup: LLM interactions do not happen in chains but rather in complex networks with various structural properties, interacting LLMs have different tasks, and humans can be in the loop, selecting specific types of content and adapting the prompt to match their expectations. Our motivation for employing this simplified set-up was two-fold: first, this experimental design is adapted from experimental studies of human and animal cultural evolution, where such simplified, abstracted settings are used to draw conclusions about real-world cultural dynamics; second,as the study of LLMs’ multi-turn dynamics remains unexplored, it appears necessary to start with a setting where causal variables can be isolated, so as to lay the foundation for incrementally improving our understanding of LLMs cultural dynamics.
> **However, to take a step toward studying more realistic settings, we followed reviewer p7tc’s suggestion of adding an additional experiment on the effect of heterogeneous chains** (i.e. where models of different families interact), that we include in the Appendix (Figure S6). In brief, we found that although the position of the attractor in heterogeneous chains is often in between the position of the homogeneous chains’ attractors, it is not systematically the case.
>
> We nevertheless fully agree with the reviewer that further work is necessary in order to build on this study and investigate the effect of additional parameters, such as more diverse tasks or more complex network structures. As indicated by the reviewer, we discuss those future directions in the Discussion (line 525).

---

> > ### Author Response · Authors · 2024-11-20
> >
> > **Part 2 / 2:**
> >
> >
> > > *The findings obtained from the work highlight the importance of understanding/evaluating LLMs dynamically have been explored in many other previous works. E.g., SOTOPIA: Interactive Evaluation for Social Intelligence in Language Agents, On the Resilience of Multi-Agent Systems with Malicious Agents.*
> >
> > We thank the reviewer for allowing us to clarify our positioning relative to other studies of LLMs’ social and collective behavior. While the two papers cited by the reviewer also focus on LLMs in social settings, the first one introduces a benchmark to evaluate social intelligence, and the second one focuses on evaluating the robustness of LLMs collectives against malicious agents. Those properties of LLMs are orthogonal to the one we study, which is the tendency of LLMs collectives to converge, at a given rate, towards attractor states. Moreover, although those papers indeed highlight that studying LLMs in social contexts is insightful, they do not empirically show that single-turn evaluation is unable to predict LLMs behavior in multi-turn settings, as done in our paper.
> > Again we thank the reviewer for underlining the need to clarify this positioning, and **we now added a paragraph “Evaluation of LLMs in social contexts” in the Related Work section, where we cite the papers mentioned by the reviewer:**
> > *“‘While not focusing on the effect of multi-turn interactions, an important body of work has started to study the social cognition of LLMs. These studies revealed that GPT-4 struggles to reach human performance with respect to social commonsense reasoning and strategic communication skills ((Zhou et al., 2024)). In a similar vein, Huang et al. (2024) found that manipulating the communication structure can make LLM collectives more resilient to the introduction of malicious agents.”.*  (line 178)
> >
> >
> > > *For different kinds of tasks, have you tried different formats of the prompts? I would imagine the prompt would change quite a lot for some findings in the paper.*
> >
> > We agree that checking that our results are robust to different phrasings of the task prompts would make the paper stronger, and thank the reviewer for this suggestion. **We therefore conducted additional experiments with Llama-3-8B where each task (rephrase, take inspiration, continue) we evaluated 5 different phrasings of the instruction prompts while keeping the meaning the same.** This revealed that our results are robust to different paraphrasing of the same prompt: indeed, the interval of the standard deviations over the 5 paraphrased prompts always contains – or almost contains – the values estimated with the original prompt in the main experiment. These results have been added to the Appendix (Figure S4), and we refer to them in the Result section (line 420).
> >
> >
> >
> > Overall, we hope to have address the reviewer’s comment with these clarifications and additional experiments, and, if so, that they will consider increasing their evaluation of our paper.

---

> > > ### Comment · Reviewer_uDks · 2024-11-25
> > >
> > > Thanks for the authors' response! With the additional experiments (where models of different families interact), which I assume are still linear, I am still not convinced that the current findings could generalize well to many downstream applications. Therefore, I will keep my score for this round.

---

> > > > ### Author Response · Authors · 2024-11-26
> > > >
> > > > We thank the reviewer for their reply. We understand than despite the additional experiments (which were aimed at making the findings more robust and the set-up more realistic), **the fact that we focus on linear chains remains a concern for the reviewer.**
> > > >
> > > > In our first response, we attempted to clarify our motivations for focusing on linear chains:
> > > > **1) this set-up is used in research on human and animal cultural evolution**, and
> > > > **2) this simplified and abstracted setting is a necessary stepping stone** for understanding LLM multi-turn dynamics in all their complexity.
> > > >
> > > > (1) **The relevance of transmission chain experiments to study the effect of iterated transmissions has been has been thoroughly advocated by scholars of cultural evolution (e.g. Mesoudi & Whiten, 2008, Derex, 2024)**. For instance, Mesoudi & Whiten (2008) argue that an “ *advantage of adopting a cultural evolutionary approach to cultural transmission is that it encourages links to be made between small-scale transmission processes that can be observed in a restricted number of individuals, as typically studied in experiments, and population-level patterns generated by people in real-life situations over longer time periods.* »
> > > > Linking macro-evolutionary patterns to small-scale processes is not specific to cultural evolution research, as it was for instance the basis of the evolutionary synthesis in the 1930s (Mesoudi, 2007).
> > > >
> > > > The assumption is therefore that « *the forces and biases of cultural transmission studied experimentally in the laboratory can be seen as at least partly generating the population-level patterns of cultural change* » (Mesoudi et Whiten, 2008, p. 3490). This assumption had been confirmed by comparing experimental results to outcomes of observational and ethnographic studies (Mesoudi & Whiten, 2008).
> > > > **We were thus building on a consensual and well-established paradigm when deciding to rely on linear transmission chains.**
> > > >
> > > >
> > > > (2) We now detail why we believe this simplified and abstracted setting is a **necessary stepping stone** for understanding LLM multi-turn dynamics in all their complexity.
> > > >
> > > > First, we agree that applying the methods and concepts we introduce to LLMs embedded in more complex network structures is a necessary next step that we would like to explore. However, it would require a dedicated study. Indeed, **moving from linear chains to complex networks would introduce several additional causal variables**, such as the size of the network, its structural properties (e.g. density, clustering) and the instructions given to LLMs for integrating information from different sources (e.g. “select one”, “combine all”, etc). Each of these variables are bound to affect attraction dynamics, and **evaluating their effect in a controlled way would require an important number of experiments.**
> > > >
> > > > Overall, we would like to emphasize that **the current paper is necessary for paving the way to such future studies**: at the methodological level, **it is unclear how could these additional variables be evaluated without the concepts and methods that we introduced in this paper.** As for the empirical results, identifying causal effects in isolation from other parameters is crucial for disentangling the role of interacting variables (Derex, 2024). For instance, we found that unconstrained tasks lead to stronger attractors; this result may then be evaluated when considering complex networks, for instance to evaluate whether specific “information aggregation” instructions or specific network structures mitigate or magnify this effect.
> > > >
> > > > We would be very keen to know whether the reviewer found these arguments convincing and, if not, why. This would allow us to potentially clarify the reasons underlying the choice of this experimental design.
> > > >
> > > > Additionally, we would like to highlight that **the robustness check recommended by the reviewer revealed that our results hold when varying the phrasing of the prompt**. We thank the reviewer for this suggestion as it greatly strengthen our results.
> > > >
> > > > **References:**
> > > >
> > > > Mesoudi, A. (2007). Using the methods of experimental social psychology to study cultural evolution. *Journal of Social, Evolutionary, and Cultural Psychology, 1*(2), 35 58. https://doi.org/10.1037/h0099359
> > > >
> > > > Mesoudi, A., & Whiten, A. (2008). The multiple roles of cultural transmission experiments in understanding human cultural evolution. *Philosophical Transactions of the Royal Society B: Biological Sciences, 363*(1509), 3489 3501. https://doi.org/10.1098/rstb.2008.0129
> > > >
> > > > Derex, M., Edmiston, P., Lupyan, G., & Mesoudi, A. (2024). Trade-offs, control conditions, and alternative designs in the experimental study of cultural evolution. *Proceedings of the National Academy of Sciences, 121*(48), e2322886121. https://doi.org/10.1073/pnas.2322886121

---

### Official Review · Reviewer_p7Tc · 2024-11-04

**Soundness:** 3
**Presentation:** 4
**Contribution:** 3
**Rating:** 8
**Confidence:** 4

**Summary:**

The authors use the storytelling medium of the 'telephone game' as a means of examining how llm text-generations evolve in a multi-turn setting. They introduce methods of analyzing the effect of multi-turn versus single turn conversations on properties such as 'positivity, difficulty, toxicity and length' and demonstrate that some properties rapidly reach saturation, depending on the exact nature of the prompt provided to the llm (ie, continue vs rephrase)

**Strengths:**

In my view, the biggest strengths of the paper are in research question and approach. There exists an extremely large body of research in investigating the single-turn capabilities of LLMs but less so in multi-turn settings. For this, the specific aspect of how certain attributes act as 'attractors' in driving the LLMs responses is extremely relevant and the authors methodology for measuring this is well-founded in the literature. The results they show, that certain attributes are (understandably) biased towards and reach saturation fairly quickly, are definitely relevant and an important contribution to the literature.

**Weaknesses:**

The primary weaknesses of the paper are in the limited number of models and that (to my understanding) each 'conversation' was conducted by the same 'model' (described in the paper as homogenous transmission chains) without an ablation of how a heterogenous transmission chain could shift results. While doing this with a number of agents n> 2 might be excessive, I do think a 2 agent heterogenous system could be reasonable.

**Questions:**

1.) How were the tasks selected?

2.) Why were other commercial (Gemini, Claude) and open-source models (Grok, Molmo) not selected for testing?

3.) What was the authors motivation in not including heterogenous transmission chains?

---

> ### Author Response · Authors · 2024-11-20
>
> We thank the reviewer for their encouraging and thorough review. We are pleased that the reviewer acknowledged the necessity to investigate LLMs capacities in multi-turn settings, and that they found the concept of attractors to be extremely relevant and our methodology well grounded.
>
> We address each of their comments below:
>
>
>
> > *How were the tasks selected?*
>
> We thank the reviewer for the opportunity to clarify how the tasks were chosen. In the submitted manuscript, we explain that these three tasks cover “typical uses of LLMs“ (e.g. Pratviel, Esteban. « Baromètre 2024 -  Les Français et les IA génératives - Vague 2 », 2024.), which are content summarization/re-writing (rephrase), creative writing (take inspiration) and dialogue generation (continue) (lines 205-214). However, we acknowledge that these tasks do not cover the full range of common use-cases. As the space of potential tasks is infinite, we decided to restrict our study to three representative tasks.
> However, following suggestions from reviewers Kc4v, 1ZZ9 and uDKs, **we added new experiments to the revised manuscript, where we manipulate the phrasing of the instruction prompts (Figure S4), and evaluate the consequences of tasks with various degree of constraints (Figure S5).**
>
>
>
>
> > *Why were other commercial (Gemini, Claude) and open-source models (Grok, Molmo) not selected for testing?*
>
> As stated by the reviewer when reviewing the strengths of the paper, the main contribution of our work is to identify and empirically show the existence of a gap in the literature (i.e. the fact that single-turn evaluations are not suited to predict the outcome of multi-turn interactions) and to provide the conceptual and methodological tools for filling this gap. As a consequence, the ambition of this work was not to provide an evaluation of all existing models, but rather to illustrate how the tools we introduce allow to perform comparison between models that have different sizes, belong to different families, are either open-source or private, and are either base or fine-tuned versions. To do so in a controlled way (changing only one parameter at a time), we included models of the same family but different size, of the same size but different families, and base versus instruct versions of the same model.
> Nevertheless, we acknowledge that including at least one recent model would make our paper more relevant to the eyes of several researchers, and **we therefore conducted additional analyses with GPT-4o-mini (OpenAI, 2024).** Indeed, it's likely that this model is and will be popular, as it's inexpensive but high-performance. We observe that using GPT-4o-mini leads to weaker attractors than GPT3.5, Llama3-8B and Mixtral-8x7B. We discuss it in more detailed in the revised manuscript (lines 413), and we updated all figures to include the results for GPT-4o-mini (Figures 3, 4 and 5).
>
> > *What was the authors motivation in not including heterogenous transmission chains?*
>
> As the study of LLMs’ multi-turn dynamics remains unexplored, it appeared necessary to start with a setting where causal variables can be isolated, so as to lay the foundation for incrementally improving our understanding of LLMs cultural dynamics. As a consequence, we focused on homogeneous chains where the multi-turn dynamics of different models can be assessed in isolation.
> However, we acknowledge that including a study on the effect of heterogeneous transmission chains would make even more salient the scope of our methodological contributions.**We therefore also include the results of this additional experiment in the revised manuscript (Figure S6). We compared 4 conditions: homogeneous chains of Mistral-7B models, homogeneous chains of Llama-3-8B models, heterogeneous chains of Mistral-7B and  Llama-3-8B models starting with Mistral-7B and heterogeneous chains of Mistral-7B and  Llama-8B models starting with Llama-3-8B.**
> In brief, we found that although the position of the attractor in heterogeneous chains is often in between the position of the homogeneous chains’ attractors, it is not systematically the case. We interpret this result more extensively in Appendix of the revised manuscript (line 1406), and refer to it in the Results section (line 314).
>
> Overall, we hope to have address the reviewer’s comment with these clarifications and additional experiments, and, if so, that they will consider increasing their evaluation of our paper.

---

### Official Review · Reviewer_1ZZ9 · 2024-11-04

**Soundness:** 3
**Presentation:** 2
**Contribution:** 3
**Rating:** 8
**Confidence:** 4

**Summary:**

This paper shows that LLMs can amplify small biases through iterative interactions. The authors show that biases, such as toxicity, can become reinforced in transmission chains. Additionally, open-ended instructions generate stronger attraction effects than constrained tasks.

**Strengths:**

The research is grounded in a strong foundation of social science theory, specifically cultural attraction theory (CAT). The phenomenon they observe is novel and interesting, and the writing is clear.

**Weaknesses:**

- The paper does not explicitly provide theories or mechanisms underlying this phenomenon. Why do LLMs exhibit this human-like behavior of transmitting cultural patterns? Do LLMs simply mimic human behavioral patterns present in the training data, or does this behavior emerge due to specific objectives in LLM training? The authors may consider discussing this further or suggesting directions for future research.

- Page 17, line 866: The paper relies on a small sample of initial texts (5 abstracts, 10 news articles, and 5 social media comments from online datasets). I am concerned about whether findings derived from such a small sample can be generalized across a broader range of tasks. The authors may consider including the full set of initial texts in the appendix so that readers can evaluate them directly. The authors may also consider replicating their main findings using paraphrased prompts to demonstrate that their results are not sensitive to prompt design.

- Page 10, line 539: Here, the authors identify the absence of network-based approaches as a limitation and an area for future work. It could be useful to suggest one or two specific network-based strategies to reduce toxic transmission and promote prosocial behaviors (e.g., lower toxicity, kindness). For instance, this paper suggests that interaction within broader social networks might reduce the transmission of toxic behaviors among LLMs: https://arxiv.org/abs/2402.12590.

**Questions:**

- Page 16, line 866: How do the authors justify the generalizability of their findings based on 20 conveniently sampled initial texts? The authors may consider providing more information in the appendix about how these texts were sampled or discussing the limitations of this sampling approach. For example, are these texts a balanced sample? (e.g., do the 5 abstracts cover a variety of scientific topics, do the 10 news articles address diverse issues, and do the 5 social media comments represent a range of user demographics?)

- Page 7, line 358: The authors conduct Kolmogorov–Smirnov (KS) tests and calculate p-values repeatedly to compare property distributions after one interaction and after multiple interactions. Given the multiple comparisons, should the authors consider applying a Bonferroni or FDR correction to control for potential false positives? Please address this by applying the appropriate correction (if applicable) or providing a justification if these adjustments are unnecessary.

- Page 5, line 237: Why did you use the term "positivity bias" instead of "negativity bias" here? Typically, "positivity bias" refers to favoring more positive words (https://www.pnas.org/doi/abs/10.1073/pnas.1411678112). However, in this context, it appears you are referring to a bias toward more negative words.

**Details Of Ethics Concerns:**

This paper discusses the amplification of small biases through iterative interactions. I believe that this method would be useful for understanding and mitigating potential risks associated with LLMs, but the topic might require ethical review.

---

> ### Author Response · Authors · 2024-11-20
>
> *Due to the character limit on OpenReview, our response will be posted in three parts.*
>
> **Part 1 / 3:**
>
> We thank the reviewer for their positive and thorough review. We are encouraged by the fact that they found the phenomenon studied interesting and novel, the writing clear, and that they acknowledge the strong theoretical grounding in cultural evolution theory.
>
> We address each of their comments below.
>
>
> > *The paper does not explicitly provide theories or mechanisms underlying this phenomenon. Why do LLMs exhibit this human-like behavior of transmitting cultural patterns? Do LLMs simply mimic human behavioral patterns present in the training data, or does this behavior emerge due to specific objectives in LLM training? The authors may consider discussing this further or suggesting directions for future research.*
>
> We thank the reviewer for this insightful comment. Investigating the mechanisms underlying the phenomena we observed is a very interesting and important direction. In the submitted manuscript, we begin to investigate this question by comparing the strength and position of attractors for models of different sizes, and find that size does not seem to impact strength or position (Figure 4). We also compare attractor properties before and after fine-tuning (Figure 5.b; line 429). This provides a first insight into the respective contribution of training data and alignment with human preferences during fine-tuning. However, we agree that this research direction is important and deserves more extensive discussion. **We expand our discussion section accordingly. In particular, we now propose studying the effect of data curation and different fine-tuning methods on attractors as an important future direction.**

---

> > ### Author Response · Authors · 2024-11-20
> >
> > **Part 2 / 3:**
> >
> > > *Page 17, line 866: The paper relies on a small sample of initial texts (5 abstracts, 10 news articles, and 5 social media comments from online datasets). I am concerned about whether findings derived from such a small sample can be generalized across a broader range of tasks. The authors may consider including the full set of initial texts in the appendix so that readers can evaluate them directly. The authors may also consider replicating their main findings using paraphrased prompts to demonstrate that their results are not sensitive to prompt design.*
> >
> > Given its length, the full set of initial texts is made available on our GitHub repository rather than in the Appendix.
> > As for the sample size, the motivation behind the choice of a rather small dataset was to show that our methods do not require extensive amounts of data in order to quantify attraction dynamics. Indeed, the current sample size (which already requires 5000 queries per model and task) already enables the detection of significant and consistent differences in attraction strength and position (e.g. for different tasks, or different text properties). However, as some of our results have interesting implications, we agree with the reviewers that verifying that they hold when using a larger sampling size would greatly improve the paper. **We therefore ran additional experiments with Llama-3-8B using 100 different initial texts** (as mentioned in the response to reviewer Kc4v who made the same comment). We observe that the trends remain the same with 100 initial texts as with 20 initial texts, and that the standard deviation intervals for 100 initial texts include – or are very close to including – the values estimated with 20 initial texts (with only one exception out of 24 values). The figure displaying those results has been added to the Appendix (Figure S3), and we refer to it in the result section (lines 420) in the revised manuscript.
> > Similarly, we agree that checking that our results are robust to different phrasings of the task prompts is important. **We therefore conducted additional experiments with Llama-3-8B where each task (rephrase, take inspiration, continue) we evaluated 5 different phrasings of the instruction prompts while keeping the meaning the same.** This revealed that our results are robust to different paraphrasing of the same prompt: indeed, the interval of the standard deviations over the 5 paraphrased prompts always contains – or almost contains – the values estimated with the original prompt in the main experiment. These results have been added to the Appendix (Figure S4), and we refer to them in the Result section (lines 420).
> >
> >
> >
> > > *Page 10, line 539: Here, the authors identify the absence of network-based approaches as a limitation and an area for future work. It could be useful to suggest one or two specific network-based strategies to reduce toxic transmission and promote prosocial behaviors (e.g., lower toxicity, kindness). For instance, this paper suggests that interaction within broader social networks might reduce the transmission of toxic behaviors among LLMs: https://arxiv.org/abs/2402.12590.*
> >
> > We thank the reviewer for this very relevant suggestion. It would indeed be very interesting to use the method we introduce to evaluate whether some network-based strategies can steer attraction dynamics in beneficial directions, such as more pro-social behavior or more diverse content. We therefore **modified the sentence at line 531 as follows:**
> > “Following some initial endeavors (Nisioti et al., 2022; Perez et al., 2024; Lai et al, 2024), future work may assess similar effects in machine networks, and in particular investigate how attraction dynamics are shaped by network-based strategies such as dynamic network structures (Nisioti et al., 2022) or free-formed decentralized networks (Lai et al, 2024).”

---

> > > ### Author Response · Authors · 2024-11-20
> > >
> > > **Part 3 / 3:**
> > >
> > > > *Page 16, line 866: How do the authors justify the generalizability of their findings based on 20 conveniently sampled initial texts? The authors may consider providing more information in the appendix about how these texts were sampled or discussing the limitations of this sampling approach. For example, are these texts a balanced sample? (e.g., do the 5 abstracts cover a variety of scientific topics, do the 10 news articles address diverse issues, and do the 5 social media comments represent a range of user demographics?)*
> > >
> > > As described above (in part 2/3), **we conducted additional experiments which confirmed that our results hold when using a larger set of initial texts.**
> > >
> > >
> > > > *Page 7, line 358: The authors conduct Kolmogorov–Smirnov (KS) tests and calculate p-values repeatedly to compare property distributions after one interaction and after multiple interactions. Given the multiple comparisons, should the authors consider applying a Bonferroni or FDR correction to control for potential false positives? Please address this by applying the appropriate correction (if applicable) or providing a justification if these adjustments are unnecessary.*
> > >
> > > We thank the reviewer for raising our attention about the potential need to control for repeated p-values computations. Controlling for the false discovery rate (FDR) using methods such as Bonferroni correction is necessary in cases where many statistical tests are conducted, as it increases the likelihood that at least one of them is found significant “by chance” (Family-Wise Error Rate; Colas, 2022). With respect to our set-up, this would have been necessary if we concluded that multi-turn interactions lead to significant differences when at least one of the 49 p-values (one by generation) was below the significance threshold. However, we rather report all p-values, and draw conclusions from the consistent trend observed over all generations, rather than from single p-values. Therefore, if we observe a general trend of decreasing p-values, we can conclude that observing the given distribution under the null hypothesis becomes less and less likely with generations. **It thus seems to us that FDR corrections are unnecessary for drawing this conclusion.** We hope this justification addresses your concern. **We clarified this point in the Method section (line 363)**(*“across most instances, we observe that the p-values steadily decrease, indicating that observing the given distribution under the null hypothesis becomes less and less likely with generations”*), and **added the more extensive justification in the Appendix (line 1462) of the revised manuscript.**
> > >
> > > >*Page 5, line 237: Why did you use the term "positivity bias" instead of "negativity bias" here? Typically, "positivity bias" refers to favoring more positive words (https://www.pnas.org/doi/abs/10.1073/pnas.1411678112). However, in this context, it appears you are referring to a bias toward more negative words.*
> > >
> > > We thank the reviewer for catching this unclear phrasing. We used the term “positivity bias” to refer to “a bias with respect to positivity”, but indeed in this context the bias is toward negative content. **We thus agree that this phrasing was somewhat confusing, and we replaced it by “negativity bias” in the revised manuscript.**
> > >
> > >
> > > > *Flag For Ethics Review:Yes, Discrimination / bias / fairness concerns, Yes, Potentially harmful insights, methodologies and applications.
> > > Details Of Ethics Concerns:
> > > This paper discusses the amplification of small biases through iterative interactions. I believe that this method would be useful for understanding and mitigating potential risks associated with LLMs, but the topic might require ethical review.*
> > >
> > > We kindly invite the reviewer to clarify their motivation for flagging our submission for ethics review. In particular, it would be useful if they could express which aspect of the ICLR code of ethics was not adequately addressed: ( https://iclr.cc/public/CodeOfEthics ). It is important for us to ensure that we align with all the listed requirements.
> > >
> > >
> > > Overall, we hope to have address the reviewer’s comment with these clarifications and additional experiments, and, if so, that they will consider increasing their evaluation of our paper.

---

> ### Comment · Reviewer_1ZZ9 · 2024-11-26
>
> I thank the authors for their detailed responses and have updated my score accordingly.
>
> Regarding the ethics review, I believe that the amplification of toxicity may unintentionally contribute to societal harms (‘unintentional harm’ in the Code of Ethics). I hope the authors will highlight the importance of studying this topic in open, scientific communities and emphasize that the responsibility lies with end-users who develop multi-turn interaction systems, possibly using ethics statement.

---

> > ### Author Response · Authors · 2024-11-26
> >
> > We would like the thank the reviewer for **raising their grade from 6 to 8** and for acknowledging that the paper was **strengthened by the additional experiments and clarifications made during the rebuttal**. The questions and suggestions of the reviewer were highly relevant and allowed us to make the paper much more compelling.
> >
> > We also thank the reviewer for clarifying their comment regarding the ethic review. We fully agree with the reviewer, and **have added the following “Ethics Statement” at the end of the revised manuscript:**
> >
> > **Ethics statement**
> >
> > *The results presented in this paper highlight that multi-turn interactions can give rise to potential harmful biases that cannot be detected by only studying the outcome of single-turn interactions. By introducing novel conceptual and methodological tools, we provide ways of quantifying the properties of multi-turn interaction systems with respect to the extent to which they magnify or reduce these biases. It is then the responsibility of end-users to make an ethical use of these tools, for instance by using them to ensure that LLMs remain aligned with ethical standards even after repeated interactions.*
> >
> > The revised manuscript including this Ethics Statement has been uploaded to OpenReview. We hope this matches the expectations of the reviewer, and remain open to making adjustments if they deem it necessary.

---

> > > ### Comment · Reviewer_1ZZ9 · 2024-11-26
> > >
> > > Thank you for the response! I think the Ethics statement thoughtfully addresses the concern I mentioned.

---

### Official Review · Reviewer_Kc4v · 2024-11-06

**Soundness:** 1
**Presentation:** 2
**Contribution:** 2
**Rating:** 3
**Confidence:** 5

**Summary:**

The paper proposes the evaluation of LLMs through doing an iterated task many times, and evaluating the subsequent effects on toxicity, positivity, difficulty, and length. The authors also evaluate the strength and position of attractors (essentially equilibria). The claims of the paper are mainly that more open-ended instructions lead to stronger attraction effects than more constrained tasks, and that some metrics have different sensitivity to attraction compared to others.

**Strengths:**

Originality: Multi-turn interactions in LLMs are an interesting an less studied area in LLM evaluations on qualities such as bias. The telephone game is an easy to grasp concept that allows readers to conceptualize the experiments easily.

Quality: The evaluations are conducted on both open- and closed-source models. The measurement methods of metrics are implemented reasonably.

Clarity: Overall the paper is easy to follow. Graphs are colorful and visually appealing.

Significance: The motivation of the paper, to study the effects of multi-step interactions of models, is meaningful to study.

**Weaknesses:**

Originality: None

Quality:
- There is little justification for why "rephrase", "take inspiration from", and "continue" are chosen as the three tasks for the models to iterate on.
- I don't think one of the main conclusions that the authors claim is supported by their experiments. The authors use these three tasks to make the claim that tasks that are less constrained have higher attractor strength, but the three conditions are difference in many different ways, not just the level of constraint; to accurately test this hypothesis, you would need a more constrained and less constrained version of the same task, such as "rephrase but start with the same first five words" versus "rephrase". I worry that there are many confounding factors (such as rephrase ping-ponging back and forth across a small set) which could lead to the lower attractor strength that the authors observe. Thus, I would hope for a more rigorous analysis with more task conditions in order to confirm the findings of the paper.
- Models are outdated, which is especially important to get right for evaluation-type papers in order to gain actionable insights.
- Sample size is very small (20 initial texts) and the domains that are chosen from are seemingly random (appendix A). This is to the point that I would question the generalizability of the results in the paper.
- Other findings, such as toxicity quickly converging to 0, is uninteresting due to safety tuning of models.

Clarity:
- Missing the entire literature on Model Collapse which is highly relevant (e.g., https://www.nature.com/articles/s41586-024-07566-y, https://openreview.net/forum?id=5B2K4LRgmz#discussion)
- Missing papers on bias in LLMs (e.g., https://arxiv.org/abs/2402.04105)
- Missing a large amount of more prominent works on models and humans displaying similar cognitive biases (line 49-50) e.g., https://arxiv.org/abs/2402.18225, https://arxiv.org/abs/2406.17055, https://www.pnas.org/doi/10.1073/pnas.2218523120
- Unfinished sentence at line 199-200: "texts spanned various types of content: scientific abstracts"
- Figure and graph text is too small, error bars are not shown.

Significance:
- I think a big issue of the paper is that it doesn't motivate the research questions. It is simply stated that multi-turn paradigms (such as scientific reviews and multi-agent environments) are common applications of LLMs, and makes the logical jump that it is necessary to evaluate these models in iterative scenarios, specifically rephrasing, inspiration, and continuation. The issue here is two-fold. First, there is little motivation in the introduction about how real-life tasks (where there are actual implications) can be adequately abstracted into these multi-turn iterative settings without losing key details that make those tasks important. For example, when news articles are continuously rephrased, often this is with human oversight --- why can this step be taken away in the experiments and what is lost from this? There does not seem to be any discussion in this direction. Next, there needs to be some generalizability statement saying that xxx are the most common use cases of LLMs in iterative settings and therefore we do rephrasing and other tasks. Without the motivation from actual use cases, it is hard to find significance in just directly analyzing iterative generation processes in LLMs because somewhat similar paradigms are used in practice.

**Questions:**

See weaknesses.

---

> ### Author Response · Authors · 2024-11-20
>
> *Due to the character limit on OpenReview, our response will be posted in three parts.*
>
> **Part 1 / 3:**
>
> We thank reviewer Kc4v for their thorough review and insightful comments. We are encouraged by the fact that they found the topic of multi-turn interaction meaningful, interesting, and currently understudied, and that they found the paper easy to follow.
>
> It appeared to us that **several of the reviewer’s comments stem from the fact that they understood this study as an “evaluation” paper (in the sense of a benchmark), while we rather aimed at providing novel ways of evaluating LLMs, by conceptualizing new, useful properties and methods to measure them.**
>
> As a general response, we would therefore like to emphasize what we see as the main contributions of our work:
> - We propose that there is a **gap in current LLM evaluations methods** (single-turn evaluations might not be suited to assess the properties of multi-turn interactions)
> - We **empirically confirm this hypothesis** by showing that multi-turn interactions indeed often lead to distributions of text properties that are significantly different from what is observed after a single interaction.
> - We **introduce novel conceptual and methodological tools to fill this gap**, grounded in research in cultural evolution.
> - We **showcase the potential of this method** by applying it to compare the **effect of different tasks, of different models, of temperature, and of fine-tuning on the properties of multi-turn interactions**.
> - As a consequence, **the ambition of this work was not to conduct an exhaustive evaluation** of state-of-the-art models on the metrics we introduced, which would have indeed required a larger set of models and a representative set of tasks.
>
> While we fully agree that using our method to explore in more detail the role that various parameters play in shaping attraction dynamics is an important direction, we believe that each of these questions should be the subject of a dedicated study. We thank the reviewer for highlighting that we may not have been precise enough about the scope of our contributions, and  **updated the “Main Contributions” paragraph with the contributions listed above.**
>
> Below, we address each of the reviewer’s comments specifically.

---

> ### Author Response · Authors · 2024-11-20
>
> **Part 2 / 3:**
>
> > *”There is little justification for why "rephrase", "take inspiration from", and "continue" are chosen as the three tasks for the models to iterate on.”*
>
> We thank the reviewer for the opportunity to clarify our choice. In the submitted manuscript, we explain that these three tasks cover “typical uses of LLMs“ (e.g. Pratviel, Esteban. « Baromètre 2024 -  Les Français et les IA génératives - Vague 2 », 2024.), which are content summarization/text simplification (rephrase), creative writing (take inspiration) and dialogue generation (continue) (lines 205-214). However, we acknowledge that these tasks are not representative of the full range of tasks people request LLMs to complete. However, our method is easily applicable to a wide range of tasks (e.g. “Generate an answer to”, “Make the text easier to understand”). As the space of potential tasks is infinite, we decided to restrict our study to three representative tasks, paving the way for future, more application-specific follow-ups.
> However, following suggestions from reviewers 1ZZ9 and uDKs, **we added new experiments to the revised manuscript, where we manipulate the phrasing of the instruction prompts (Figure S4)**. This confirmed that our results are not affected by different phrasings of the same instruction.
>
>
> > *I don't think one of the main conclusions that the authors claim is supported by their experiments. The authors use these three tasks to make the claim that tasks that are less constrained have higher attractor strength, but the three conditions are difference in many different ways, not just the level of constraint [...]*
>
> We fully agree with the reviewer that there might be other mechanisms at play, and that testing this specific hypothesis (i.e. the fact that less constrained tasks create stronger attractors) would require manipulating the level of constraint of the task while controlling for other factors. Our experimental set-up was indeed not specifically designed to test this hypothesis, and we rather provided a post-hoc interpretation of a significant and consistent tendency observed in our results. We agree with the reviewer that confirming this interpretation by conducting a more controlled experiment would be a great addition to the paper.
>
> **We therefore ran an additional experiment with Meta-Llama-3-8b where we instructed the LLM to create a new text by making either “very minor”, “a few small”, “moderate”, “notable” or “radical” changes to the received text.** This experiment confirmed our initial interpretation, as less constrained tasks indeed exhibit stronger attractors. The only exception is Length. We interpret this by the fact that going from “very minor” to “radical” changes mainly refers to the semantic of the text. As a consequence, the instruction to make “minor changes” is not more constrained than to make “radical changes” with respect to Length.
> The figure presenting these results has been added to the Appendix of the revised manuscript (Figure S3), and we refer to it in the Results section (line 420).
>
> > *Models are outdated, which is especially important to get right for evaluation-type papers in order to gain actionable insights.*
>
> As detailed in the previous comments, our primary aim was not to provide an evaluation of SOTA models, but rather to illustrate how the tools we introduce enables to map various properties of models, and how these maps of properties depend on various factors like
> sizes, model families, being either open-source or private, and being either base or fine-tuned versions. To do so in a controlled way, we included models of the same family but different size, of the same size but different families, and base versus instruct versions of the same model. Moreover, Llama-3-8b and Llama3-70b are only recently “outdated” (e.g. Llama-3.1 was released at the end of July 2024, two months before the ICLR submission deadline).
> Nevertheless, we acknowledge that including at least one recent model would be a nice addition, and **we therefore conducted additional analyses with GPT-4o-mini (OpenAI, 2024)**. Indeed, it's likely that this model is and will be popular, as it's inexpensive but high-performance. We observe that using GPT-4o-mini leads to weaker attractors than GPT3.5, Llama3-8B and Mixtral-8x7B. We discuss it in more detail in the revised manuscript (lines 413), and we updated all figures to include the results for GPT-4o-mini (Figures 3, 4 and 5).

---

> > ### Author Response · Authors · 2024-11-20
> >
> > **Part 3 / 3:**
> >
> > > *Sample size is very small (20 initial texts) and the domains that are chosen from are seemingly random[...]*
> >
> > The motivation behind the choice of a rather small dataset was to show that our methods do not require extensive amounts of data to quantify attraction dynamics. Indeed, this small sample size already allows us to detect statistically significant and consistent differences in attraction strength and position. However, we agree with the reviewer that verifying that our results hold when using a larger sample size would greatly improve the paper. **We therefore ran additional experiments with Meta-Llama-3-8b using 100 different initial texts.** We observe that the trends remain the same with 100 initial texts as with 20 initial texts, and that the standard deviation intervals for 100 initial texts include – or are very close to including – the values estimated with 20 initial texts (with only one exception out of 24 values). The figure displaying those results has been added to the Appendix (Figure S3), and we refer to it in the Result section (line 420) in the revised manuscript.
> >
> >
> > > *Other findings, such as toxicity quickly converging to 0, is uninteresting due to safety tuning of models.*
> >
> > We believe that the fact that this finding is not surprising does not make it uninteresting, as several papers show that some elicitation methods can reveal biases in models despite safety tuning. Before running our experiments, it was therefore still an open question whether iterated transmission would make models deviate from their safety tuning.
> > Moreover, the method we introduce allows not only for the quantification of the attractor’s position (e.g., ~0 for toxicity ), but also the strength of attraction. For instance, this allows us to observe that fine-tuning increases the strength of attraction for toxicity, but that this attractor is already quite strong in base models before fine-tuning (line 429).
> >
> > > *Missing the entire literature on Model Collapse which is highly relevant [...] on bias in LLMs [...] on models and humans displaying similar cognitive biases*
> >
> > We thank the reviewer for highlighting relevant papers we did not cite. **We have incorporated all suggestions in the Related Work section of the revised manuscript, alongside new references (Kazdan, 2024; Dohmatob, 2024)**. That said, the original manuscript already does integrate relevant literature on model collapse (line 154). We would like to highlight that the paper Shumailov et al., 2023 that we cite in the original manuscript is the pre-print version of the paper suggested by the reviewer (Shumailov, 2024) We now cite the published version instead.
> >
> >
> > > *Figure and graph text is too small, error bars are not shown.*
> >
> > Figure and graph text are now bigger in the revised manuscript. Error bars have been added to Figure 4.
> >
> >
> > > *I think a big issue of the paper is that it doesn't motivate the research questions[...] Without the motivation from actual use cases, it is hard to find significance in just directly analyzing iterative generation processes in LLMs because somewhat similar paradigms are used in practice.*
> >
> > We thank the reviewer for inviting us to comment on the ecological validity of our experimental design. The real-life dynamics of LLM content generation are indeed much more complex than our experimental setup: LLM interactions do not happen in chains but rather in complex networks with various structural properties, interacting LLMs have different tasks, and humans can be in the loop, selecting specific types of content and adapting the prompt to match their expectations. Our motivation for employing this simplified set-up was two-fold: first, **this experimental design is adapted from experimental studies of human and animal cultural evolution, where such simplified, abstracted settings are used to draw conclusions about real-world cultural dynamics**; second, as the study of LLMs’ multi-turn dynamics remains unexplored, **it appears necessary to start with a setting where causal variables can be isolated, so as to lay the foundation for incrementally improving our understanding of LLMs cultural dynamics.**
> >
> > Then, we fully agree with the reviewer that further work is necessary in order to build on this study and investigate how additional parameters, such as human supervision, affect the attraction dynamics.
> > Examples of future directions can be found in the section “Limitations and Future Directions” (e.g. “our study could also be extended to hybrid networks in which humans and LLMs interact” (line 537)).
> >
> >
> > Overall, we hope to have address the reviewer’s comment with these clarifications and additional experiments, and, if so, that they will consider increasing their evaluation of our paper.

---

> > > ### Author Response · Authors · 2024-11-27
> > >
> > > Dear Reviewer  Kc4v,
> > >
> > > As we approach the end of the rebuttal period, we notice that you have not yet responded to our detailed answers to your concerns.
> > >
> > > Your score differs substantially from other reviewers who rated the paper more favorably. Since ICLR typically requires consensus among reviewers for acceptance, your current score could prevent the paper from being published despite the positive feedback from other reviewers.
> > >
> > > We have worked diligently to address each of your concerns by:
> > >
> > > -	Clarifying the main contributions $-$ and updating the manuscript accordingly  $-$ to emphasize that the aim was not to exhaustively benchmark LLM models.
> > >
> > > -	Conducting additional experiments where we manipulate the phrasing of the instruction prompts.
> > >
> > > -	Conducting additional experiments where we specifically manipulate the level of constraints in the instruction prompts.
> > >
> > > -	Replicating all of our experiments with a more recent model (GPT-4o-mini)
> > >
> > > -	Conducting additional experiments with a larger sample of initial texts
> > >
> > > -	Updating the Related Work section with additional references
> > >
> > > -	Revising the figures to improve clarity
> > >
> > > All of these revisions have been integrated to the new version of the manuscript which has been uploaded to OpenReview.
> > >
> > > Given these improvements and our detailed responses, would you be willing to review our rebuttal and reconsider your rating?
> > >
> > > If there are remaining concerns that would need to be addressed to merit a higher score, we would greatly appreciate your feedback before the rebuttal period ends, in order to make the most out of the discussion period.

---

> ### Comment · Reviewer_Kc4v · 2024-12-03
> **Response to rebuttal**
>
> After reading all of the rebuttal and the updated paper, I still have several points of concern:
>
> "several of the reviewer’s comments stem from the fact that they understood this study as an “evaluation” paper, while we rather aimed at providing novel ways of evaluating LLMs, by conceptualizing new, useful properties and methods to measure them."
>
> I think this framing, while better supported by the authors' results, is actually weaker than the evaluation framing -- instead of showing a result that is relevant to the ML community, now the authors are simply making the point that multi-turn LLM paradigms should be studied, which isn't new to the NLP community, and thus I don't think this is a novel enough takeaway. Papers since 2017 and 2018 have tackled multi-turn dialogue settings in traditional NLP: e.g., https://arxiv.org/abs/1710.03957, https://arxiv.org/abs/1810.00278. While the authors take a specific approach of multi-turn by continually asking the model to modify the text in some way, I think this is actually less realistic than existing multi-turn benchmarks and aren't particularly more important to study.
>
> Also, upon further literature review, I don't believe the paper significantly differentiates itself from the following paper, which has been published in PNAS in 2023. With respect to this paper, the contribution of the methodology is simply not substantially novel:
> A. Acerbi and J. M. Stubbersfield. Large language models show human-like content biases in transmission chain experiments. Proceedings of the National Academy of Sciences, 120(44): e2313790120, 2023.
>
> "“typical uses of LLMs“ (e.g. Pratviel, Esteban. « Baromètre 2024 -  Les Français et les IA génératives - Vague 2 », 2024.), which are content summarization/text simplification (rephrase), creative writing (take inspiration) and dialogue generation (continue)"
>
> I don't believe this is a proper source to cite given much more prevalent literature surrounding use cases of LLMs, as well as lacking tasks that have been defined as important to study in NLP. For instance, QA (question answering) tasks are among the most studied and characterized tasks in NLP that meaningfully extend to LLMs, but the closest you have here is "continue", where the model continues a passage.
>
> "This experiment confirmed our initial interpretation, as less constrained tasks indeed exhibit stronger attractors (FigureS5). The only exception is Length. We interpret this by the fact that going from “very minor” to “radical” changes mainly refers to the semantic of the text. "
>
> With the length being an exception, I think the attractor strength depends on both the level of constraint of the task AND the metric you are using. For example, if the metric was not "length" but "level of detail" or something which is correlated with length but also has semantic connections, the author's explanations may not hold. Moreover, since there are exceptions to the pattern, it still feels that the authors have not completely captured a pattern --- and their conclusion is still prone to post-hoc fitting.
>
> Generally, I feel strongly that because of
> 1. framing that does not produce significant takeaways,
> 2. highly related work that is not properly represented in the paper,
> 3. weakly justified design claims that go against established important questions in the field, and
> 4. under-explained exceptions that make the conclusions prone to post-hoc fitting,
>
> while I do appreciate the additional experiments and efforts from the authors, despite the other reviewer's scores, I believe that this paper has fundamental issues keeping it from being at the level of an ICLR publication and recommend rejection.

---

> > ### Author Response · Authors · 2024-12-03
> >
> > We thank the reviewer for their detailed response to our rebuttal.
> >
> >
> > >*I think this framing, while better supported by the authors' results, is actually weaker than the evaluation framing -- instead of showing a result that is relevant to the ML community, now the authors are simply making the point that multi-turn LLM paradigms should be studied, which isn't new to the NLP community, and thus I don't think this is a novel enough takeaway. Papers since 2017 and 2018 have tackled multi-turn dialogue settings in traditional NLP: e.g., https://arxiv.org/abs/1710.03957, https://arxiv.org/abs/1810.00278. [...]*
> >
> > We believe saying that we are “*simply making the point that multi-turn LLM paradigms should be studied*” is a substantial under-statement of our contribution.
> >
> > First, our work empirically demonstrates that the properties of LLM outputs after multiple interactions differ significantly from those observed after a single interaction (Section 4.2 - Figure 3). To the best of our knowledge, **our paper is the first to provide this result**. Second, beyond highlighting a gap in existing evaluation methods, we then follow by providing novel concepts and methods that allow to characterize the properties of LLMs in multi-turn settings. Finally, we apply these methodological tools to study novel questions about LLMs multi-turn dynamics (e.g. effect of task, of sampling temperature, of fine-tuning).
> >
> > **Our contribution is thus orthogonal to the focus of the papers on multi-turn dialogue mentioned by the reviewer**. Specifically, those papers introduce datasets for training language models and benchmarking their ability to generate human-like dialogues. They are not aimed at studying how repeated interactions lead text properties to evolve over time. As such, they belong to a very different research area.
> >
> > >*“I don't believe the paper significantly differentiates itself from the following paper, which has been published in PNAS in 2023.“*
> >
> > While this paper (that we already cite in the Related Work section) is methodologically related, as it uses transmission chain experiments, there are several major differences. Their aim is to determine whether biases found in human transmission chain experiments are replicated when using LLMs. To this end, they focus on a single task (“*summarize*”), use chains of 3 to 5 generations, rely on manual annotations and focus on the direction in which a specific story evolves, following the approach used in the corresponding human experiments.
> > The aim of our paper is very different. Our focus is on providing and using novel methods to evaluate LLMs. Specifically, we introduce the concepts of attractor position and strength, which go beyond examining the direction in which a single text evolves. Additionally, we use these concepts to study how different parameters (e.g. different tasks and task phrasings, sampling temperature, fine-tuning) affect LLM behaviours in multi-turn settings.
> >
> >
> >
> > >*“I don't believe this is a proper source to cite given much more prevalent literature surrounding use cases of LLMs”*
> >
> > The cited study was conducted by the **French Institute of Public Opinion (IFOP)**, one of the major survey institutes in France, **making it a very reliable source** (e.g. it has consistently adhered to the ethics and guidance code of the European Society for Opinion and Marketing Research, ESOMAR). The claim of the reviewer that more prevalent sources exist is hard to evaluate as they do not provide any examples of such sources.
> >
> > >*“With the length being an exception, I think the attractor strength depends on both the level of constraint of the task AND the metric you are using.[...]”*
> >
> > We believe that **the reviewer’s interpretation of our results is fully compatible with the interpretation we present in the manuscript**. Specifically, we suggest that the attractor strength depends on the level of constraint of the task with respect to the metric of interest (line 1377). Therefore, as the reviewer rightly points out, attractor strength is shaped by the interplay between the task and the metrics. That said, we fully agree that, as with any quantitative result, our finding warrants further experiments to refine our understanding of the phenomenon.
> >
> > More generally, the fact that the reviewer engages in a debate about what modulates the strength of attraction highlights the value of our contribution. Such a discussion would not be possible without the concepts and methods introduced in our work. In particular, the concept of attractor strength did not exist prior to our work, which clearly demonstrates the added value of our paper (e.g. in comparison to Acerbi & Stubbersfield, 2023).
> > **We therefore view the reviewer’s comment as a confirmation that our paper enables the formulation and testing of hypotheses that were out of reach until now.**

---

### Author Response · Authors · 2024-11-20
**Global response to all reviewers**

We would like to start by thanking all four reviewers for their highly relevant feedback. We found it very encouraging that reviewers felt that the writing was clear and the paper easy to follow (reviewers **Kc4v** and **1ZZ9**) and that the research question was interesting, original (**Kc4v** and **1ZZ9**) and filling a gap in the current literature (**Kc4v, p7Tc, uDk**s). We are glad to see the reviewers acknowledge the strong theoretical grounding of our methodology (**1ZZ9, p7Tc**), as well as its novelty and the complexity of the analyses (**uDks**). Finally, we appreciate that reviewers recognized the relevance and importance of our contribution (**p7Tc**), and its implications for evaluating and aligning LLMs (**uDks**).

The fact that none of the reviewers identified major issues such as technical flaws, or inadequate reproducibility gives us good hope that we can adequately address all concerns.  We do so by providing clarifications about the aim of our study, and by running additional experiments. **The manuscript has been revised accordingly, and the updated version uploaded.**
We briefly summarize these revisions in this general response, and provide more detailed responses to each reviewer.

**Remarks concerning the exhaustiveness of our evaluations**

Reviewers Kc4v and p7Tc raised questions about the motivation underlying our choice to focus on three specific tasks, and uDKs indicated that those tasks did not seem diverse and realistic enough. Kc4v and p7Tc asked why we did not choose to include other and more recent LLM models. Finally, p7Tc and uDKs questioned our choice to focus on linear homogeneous chains (i.e. composed of one type of model) rather than heterogeneous chains or networks.

As a general response, we would like to emphasize what we see as the main contributions of our work:
- We propose that there might be a **gap in current LLM evaluations methods** (single-turn evaluations might not be suited to assess the properties of multi-turn interactions)
- We **empirically confirm this hypothesis** by showing that multi-turn interactions indeed often lead to text distributions that are significantly different from what is observed after a single interaction.
- We **introduce novel conceptual and methodological tools to fill this gap**, grounded in research in cultural evolution.
- We **showcase the potential of this method** by applying it to compare the **effect of different tasks** (representative of common use cases of LLMs), **different models, of temperature, and of fine-tuning on the properties of multi-turn interactions**.
- As a consequence, **the ambition of this work was not to exhaustively benchmark state-of-the-art models** on the metrics we introduced.

Nevertheless, we acknowledge that including a very recent model would be a nice addition, and **we therefore conducted additional experiments with GPT-4o-mini (OpenAI, 2024) that we include in the revised manuscript**. We observe that using GPT-4o-mini leads to weaker attractors compared to other models, and further discuss it in the revised manuscript. If needed, we could easily add results on other models.

We also ran **additional experiments to study the effect of heterogeneous transmission chains**. We found that **although the position of the attractor in heterogeneous chains is often in between the position of the homogeneous chains’ attractors, it is not systematically the case**. We interpret this result more extensively in the revised manuscript.

**Remarks regarding the robustness of our results**

A second set of remarks suggested conducting additional controls and robustness checks of our results. Reviewers 1ZZ9 and Kc4v recommended verifying that our results hold when the set of initial texts is larger, and reviewer uDKs suggested evaluating the sensitivity of our metrics to different prompt phrasings. Reviewer Kc4v indicated that it might be necessary to include more controlled experiments to directly test our observation that less constrained tasks appear to create stronger attractors.

We fully agree with all of these suggestions, and believe that conducting these additional controls can only make our paper more compelling. **We therefore ran new experiments and included them in the revised manuscript**.
We found that **neither increasing the size of the set of initial texts nor paraphrasing the instructions prompt appears to have significant effects on the trends observed in the main experiment**.
As for whether less constraining tasks lead to stronger attractors, **additional experiments manipulating only this variable also confirmed the observation from the experiments, revealing that allowing for more changes indeed leads to stronger attractors.**

Overall, we believe those additional experiments and clarifications strengthen the paper, and we thank the reviewers for their providing highly relevant feedback.

---

> ### Author Response · Authors · 2024-12-03
> **Summary of the discussion period**
>
> We thank the reviewers for participating in the discussion period. We here briefly summarize the interactions that happened during this period, and refer readers to the responses to each reviewer for the complete discussions.
>
> **Reviewer 1ZZ9** acknowledged that the work we conducted during the rebuttal addressed their concerns, and **raised their grade from 6 to 8 as a consequence**. Their reply also made clearer why they believed an Ethics Statement would be necessary, and we therefore added this statement at the end of the revised manuscript. Following this addition, they **further increased their confidence from 2 to 4**.
>
> **Reviewer uDks** replied by saying that despite our additional experiments (in which we ran several robustness checks following their suggestions), the fact that we rely on linear transmission chains remains a concern. **They therefore kept their grade to 5**. We responded by re-emphasizing the motivations underlying the choice of this experimental design. Briefly, these are that **1) this set-up is used in research on human and animal cultural evolution**, and **2) this simplified and abstracted setting is a necessary stepping stone** for understanding LLM multi-turn dynamics in all their complexity. See the full responses for details. Unfortunately, reviewer uDks did not respond again.
>
> **Reviewer Kc4v** also responded to our rebuttal.
> While we ran several additional experiments in order to address their concerns, those were hardly acknowledged by the reviewer. Some parts of their response raised new concerns, such as asking our positioning with respect to additional papers and research areas. **We addressed those additional remarks in our response**, but could not integrate them into the manuscript as the reviewer’s reply was made one day before the end of the discussion period (thus after the deadline for uploading a revised version).
>
> In other parts of the response, reviewer Kc4v commented on our rebuttal. One of their concerns relates to the clarification we made about our study not being an evaluation-type paper. This in their view weakens our paper, and they claim that we are thus “simply making the point that multi-turn LLM paradigms should be studied”. **As we believe this is a substantial understatement of our contribution, we reiterated the scope of our submission**.
>
> The reviewer also suggested a different interpretation of our results about the effect of constrained tasks on attractor strength. We responded by explaining why their interpretation was actually fully consistent with ours. Moreover, we argue that the very fact that the reviewer reflects on the factors that may modulate attractor strength highlights the value of our contribution. Indeed, this discussion would not be possible without the concepts and methods introduced in our work. **Thus, the reviewer’s comment clearly illustrates how our contribution opens new avenues for studying the properties of LLMs**.
> The reviewer nevertheless **kept their rating of 3.**
>
> **Reviewer p7Tc**, who initially gave rating 8 to our paper, did not answer to our rebuttal, probably because they only had minor (yet insightful) remarks that we addressed in our response. **Their score thus remained at 8**.

---

### Meta-Review · Area_Chair_r6hk · 2024-12-19

**Metareview:**

The paper draws from CAT (cultural attraction theory) to provide a way of thinking about multi turn evaluations for LLMs. It shows that there are significantly different properties of text generated by LLMs when shifting from the single to multi turn context. While the multi-turn framing isn't particularly novel for an NLP community, the insights from CAT to evaluate LLMs potentially have value.

After going carefully through all of the reviews, it seems like the major concerns are about the overall generalizability of the tasks chosen, the way the paper is framed, and to ask for additional citations in the context of relevant literature. The authors have agreed to add the additional citations. As for how the paper is framed, I tend to agree with the authors that this not an evaluation paper in the sense of a benchmark but rather providing value in thinking about multi turn evaluations. While this framing isn't entirely clear even in the updated manuscript, this is not an unfixable issue and I hope to see this further clarified in the camera ready. As for the issues regarding generalizability, the authors should make more explicit their reasons for the choices of tasks - but I agree with them and reviewer p7Tc that this are not necessarily impeding the core contributions of the work which is to introduce a new way of thinking about multi turn evaluations. Many of the questions and concerns raised by the reviewers could likely be entire paper unto themselves.

Overall, even though there are some design flaws in this study and it could be strengthened, I believe it passes the threshold to warrant publication.

**Additional Comments On Reviewer Discussion:**

I have to discount review 1ZZ9 as even though they gave the paper a score of 8, their strengths section was not very informative of why this paper should be accepted. The remaining reviewers are split 2/3 in favor of rejection, especially reviewer Kc4v but had rebuttal periods where there was extensive discussion during which I believe many, if not all, of the reviewer's questions were answered. This of course raised followup questions but overall the authors had a good faith attempt to clarify reviewer concerns.

---

### Decision · Program_Chairs · 2025-01-22

Accept (Poster)